# SRSF6 balances mitochondrial-driven innate immune outcomes through alternative splicing of BAX

Allison R Wagner, Chi G Weindel, Kelsi O West, Haley M Scott, Robert O Watson, Kristin L Patrick*

Department of Microbial Pathogenesis and Immunology, Texas A&M Health, School of Medicine, Bryan, United States

**Abstract** To mount a protective response to infection while preventing hyperinflammation, gene expression in innate immune cells must be tightly regulated. Despite the importance of pre-mRNA splicing in shaping the proteome, its role in balancing immune outcomes remains understudied. Transcriptomic analysis of murine macrophage cell lines identified Serine/Arginine Rich Splicing factor 6 (SRSF6) as a gatekeeper of mitochondrial homeostasis. SRSF6-dependent orchestration of mitochondrial health is directed in large part by alternative splicing of the pro-apoptosis pore-forming protein BAX. Loss of SRSF6 promotes accumulation of BAX-κ, a variant that sensitizes macrophages to undergo cell death and triggers upregulation of interferon stimulated genes through cGAS sensing of cytosolic mitochondrial DNA. Upon pathogen sensing, macrophages regulate SRSF6 expression to control the liberation of immunogenic mtDNA and adjust the threshold for entry into programmed cell death. This work defines BAX alternative splicing by SRSF6 as a critical node not only in mitochondrial homeostasis but also in the macrophage's response to pathogens.

## Editor's evaluation

SR proteins are a family of RNA binding proteins that have widespread essential functions throughout biology. SRSF6 is an understudied SR family member, best characterized for its role in controlling alternative splicing. Through comparative RNA-Seq analysis, this study demonstrates a role of SR protein SRSF6 in regulating interferon-responsive gene expression in macrophages. Moreover, the data provide compelling evidence that SRSF6 influences the interferon response through controlling mitochondrial damage triggered by a spliced isoform of BAX.

*For correspondence:
kpatrick03@tamu.edu

**Competing interest:** The authors declare that no competing interests exist.

## Introduction

When innate immune cells like macrophages sense pathogen or damage associated molecular patterns (PAMPs or DAMPs), they rapidly induce transcription of hundreds of genes encoding cytokines, chemokines, and antimicrobial mediators (*Hagai et al., 2018*; *Ramsey et al., 2008*). While these transcripts are being synthesized by RNA polymerase II, they are subject to several critical co-transcriptional processing steps including 5' capping, cleavage and polyadenylation, and pre-mRNA splicing, whereby introns are removed and exons are ligated together to generate mature RNAs (*Carpenter et al., 2014*). Pre-mRNA splicing plays a key role in global regulation of the transcriptome and thus the proteome, with 92–94% of the human genome subject to alternative splicing (*Wang et al., 2008*) and >80% of alternative splicing predicted to impact protein functionality (*Yura et al., 2006*).

Splicing regulatory proteins play a critical role in maintaining the fidelity of splicing while permitting the flexibility needed for alternative exon usage. One major family of splicing regulators is the

Serine/arginine rich, or SR proteins. These proteins recognize and bind to exonic splicing enhancer sequences to define exon locations, thus directing the U snRNPs to cis-splicing signals in nearby introns. The SRs also function at other steps of the RNA life cycle including mRNA export, localization, decay, and translation (*Howard and Sanford, 2015*). Many connections have been made between SR proteins and cancer, with aberrant expression of SR proteins commonly observed in patients with multiple myeloma and acute myeloid leukemia (*Liu et al., 2022*; *Song et al., 2019*; *Wan et al., 2019*). There are emerging roles for SR proteins in regulating immune homeostasis. SRSF1, the best studied SR protein, limits autoimmunity via a role in maintaining healthy regulatory T cells (*Katsuyama and Moulton, 2021*). SRSF3 has been shown to negatively regulate IL-1β release during *Escherichia coli* infection of THP-1 monocytes (*Moura-Alves et al., 2011*) and SRSF2 promotes herpes simplex virus replication by binding to viral promoters and controlling splicing of viral transcripts (*Wang et al., 2016*).

To better define the contributions of SRSF proteins to innate immunity, we carried out a transcriptomics study of a panel of SRSF knockdown (KD) RAW 264.7 macrophage cell lines. This analysis revealed a remarkable degree of diversity in how individual SR proteins influence innate immune responses (*Wagner et al., 2021*). One particularly striking phenotype we uncovered was that of *Srsf6* KD macrophages, which express high basal levels of *Ifnb1* and interferon stimulated genes (ISGs). SRSF6 is a 55 kDa protein that is essential for viability of *Drosophilia melanogaster* and *Mus musculus* (*Mason et al., 2020*; *Ring and Lis, 1994*). Multiple studies report that SRSF6 preferentially binds purine-rich exonic splicing enhancers, with a predicted consensus site in humans of USCGKM (where S represents G or C; K represents U or G; M represents A or C) (*Liu et al., 1998*). SRSF6 activity is at least in part controlled via phosphorylation by the dual specificity kinases CLK1 and DYRK1a (*Hara et al., 2013*; *Yin et al., 2012*), which activates SRSF6 shuttling between the cytoplasm and the nucleus (*Sapra et al., 2009*). SRSF6 abundance has been repeatedly associated with cancer, liver disease, and diabetes (*Jensen et al., 2014*; *Juan-Mateu et al., 2018*; *Li et al., 2021*) and recent work has identified several roles for SRSF6 in mitochondrial function and cell death. For example, loss of SRSF6 decreases mitochondrial respiration, leading to increased cellular apoptosis in human endoC-BH1 endothelial cells, likely via alternative splicing of cell death factors like *Bim*, *Bax*, *Diablo*, and *Bclaf1* (*Juan-Mateu et al., 2018*). Likewise, phosphorylation of SRSF6 induces alternative splicing of mitochondria related genes (e.g. *Polg2*, *Nudt13*, *Guf1*, *RnaseI*, and *Nme4*) in a mouse model of fatty liver disease as well as in human hepatitis patients (*Li et al., 2021*).

Here, we report that SRSF6 works to limit basal type I interferon (IFN) expression and apoptosis in murine macrophage cell lines and primary macrophages. The mechanisms underlying these phenotypes converge on alternative splicing of the pro-apoptotic factor BAX; specifically, upregulation of a BAX variant called BAX-kappa (Bax-κ). Our findings support a model whereby Bax-κ expression renders BAX mitochondrial pores permissive to mtDNA release and sensitive to triggers of programmed cell death. These studies provide insight into how BAX alternative splicing controls mitochondrial homeostasis and illuminate an unappreciated role for SRSF6 in balancing macrophage innate immune responses.

## Results

### SRSF6 knockdown activates type I interferon gene expression in macrophages

To appreciate the distinct contribution of individual SR proteins to macrophage gene expression, we generated RAW 264.7 macrophage cell lines (RAW MΦ) stably expressing shRNA hairpins directed against *Srsf*1, 2, 6, 7, and 9, as previously reported in *Wagner et al., 2021*. Most SR proteins are ubiquitously expressed across cell types, and we confirmed expression of each of these SRs in our RAW MΦ (*Figure 1—figure supplement 1A*). Because it is well-established that several members of the SR protein family are essential genes across multiple cell types (*Feng et al., 2009*; *Goldberger et al., 2021*; *Ortiz-Sánchez et al., 2019*; *Wang et al., 2001*; *Xu et al., 2005*) and because high-quality transcriptomics analyses have been carried out successfully for SR family members using shRNA or siRNA gene silencing (*ENCODE Project Consortium, 2004*; *Van Nostrand et al., 2020*), we opted to stably knockdown these factors, instead of relying on CRISPR-mediated gene editing. To identify major transcriptomic changes due to loss of SR proteins, we isolated total RNA from each of these KD

cell lines and a scramble (SCR) control that was selected alongside the SR KDs, performed RNA-seq, and measured differential gene expression using the CLC Genomics Workbench, as in *Wagner et al., 2021*. Using a simple hierarchical clustering algorithm, we visualized the gene expression profiles of each SR KD macrophage cell line and pinpointed SRSF6 as unique amongst the SR proteins queried (correlation between SRSF6's node and the rest of the tree = 0.138) (*Figure 1A*). Manual annotation of genes in clusters that were uniquely impacted by loss of SRSF6 revealed several downregulated genes related to mitochondrial biology (*Figure 1B*). On the other hand, many upregulated genes fell into the category of interferon stimulated genes (ISGs) (*Figure 1C*), as defined by *Kim et al., 2018*; *Liu et al., 2012*; *Thomas et al., 2006*; *Zahoor et al., 2014*. ISGs are a group of genes whose transcription is activated through IFNAR receptor-mediated type I interferon signaling. Generally, very few ISG transcripts accumulate in resting macrophages. Direct visualization of RNA-seq reads using the Integrated Genome Viewer shows elevated reads across all coding exons for representative ISGs *Rsad2* (*Figure 1D*) and *Mx1* (*Figure 1—figure supplement 1B*). Ingenuity Pathway Analysis confirmed overrepresentation of differentially expressed genes in functional categories related to interferon and antiviral responses (*Figure 1E*), indicating an overall increase in type I IFN signaling. Furthermore, RT-qPCR confirmed basal ISG expression in two independently derived *Srsf6* KD cell lines (*Figure 1H–J*), with the degree of ISG accumulation correlating with SRSF6 KD efficiency and protein expression (*Figure 1F–G*). We also measured ISG accumulation upon transient siRNA KD of SRSF6 in RAW MΦ cells (*Figure 1—figure supplement 1C*), arguing against off-target effects resulting from stable selection of KD cell lines. Importantly, this phenotype was recapitulated by *Srsf6* siRNA KD in primary cell types including bone marrow derived macrophages (BMDMs) (*Figure 1K*) and mouse embryonic fibroblasts (MEFs) (*Figure 1—figure supplement 1D*). Differential expression of genes involved in type I IFN responses was not observed in other resting SRSF KD RAW MΦ cell lines, suggesting this phenotype is unique to loss of SRSF6 and is not a general consequence of interfering with splicing (*Wagner et al., 2021*).

Having concluded that SRSF6 plays a role in regulating basal ISG expression, we sought to investigate the cell-intrinsic vs. cell-extrinsic nature of this phenotype. Phosphorylation of IFN regulatory factor 3 (IRF3) is a critical step in the initial stages of PAMP and DAMP sensing that lead to production of IFN-β. Immunoblot analysis revealed increased levels of phospho-IRF3 (S396) in resting *Srsf6* KD macrophages (*Figure 1L*). Higher levels of IFN-β protein were also measured in the supernatants of *Srsf6* KD RAW MΦ cells via ISRE reporter cells (*Hoffpauir et al., 2020*; *Figure 1M*). Consistent with higher levels of IFN-β secretion, supernatants from *Srsf6* KD macrophages were sufficient to stimulate ISG expression in naïve wild-type RAW MΦ cells (24 hr incubation) (*Figure 1N*). Elevated basal ISG expression in *Srsf6* KD cells was rescued by treatment with an IFN-β neutralizing antibody (*Figure 1O*). Together, these data suggest that IRF3-mediated expression of IFN-β drives upregulation of basal ISG expression in *Srsf6* KD RAW MΦ cells. Finally, to demonstrate the biological relevance of increased SRSF6-dependent basal ISG expression, we infected cells with the single stranded RNA virus VSV, which is hypersensitive to even low levels of ISG expression (*Wagner et al., 2021*; *West et al., 2019*). We observed a dramatic restriction of VSV replication in *Srsf6* KD RAW MΦ cells compared with SCR controls (*Figure 1P* and *Figure 1—figure supplement 1E*), supporting a *bona fide* role for SRSF6 as a negative regulator of antiviral immunity.

## Cytosolic mtDNA triggers cGAS-dependent DNA sensing in *Srsf6 KD* macrophages

We next sought to identify the trigger of the higher basal type I IFN expression in *Srsf6* KD RAW MΦ cells. Mitochondrial DNA (mtDNA) can activate cytosolic DNA sensing pathways when mitochondria are damaged or depolarized such that mtDNA is released from the inner matrix (*West et al., 2015*). Because mitochondrial genes were downregulated in *Srsf6* KD RAW MΦ (*Figure 1B*), we hypothesized that the mitochondrial network might be impacted by a loss of SRSF6. To begin to establish mtDNA as a source of the increased ISG expression, we first set out to determine if cytosolic mtDNA was increased in *Srsf6 KD* macrophages. To measure cytosolic mtDNA we used differential centrifugation to separate cytosolic and cellular membrane fractions (the latter of which include mitochondria) from SCR and *Srsf6* KD RAW MΦ (*Figure 2A*). We detected significant enrichment of several mtDNA genes (*Cytb*, *Dloop2*, and *Dloop1*) in the cytosol of *Srsf6* KD RAW MΦ cells (*Figure 2B*). Additionally, we determined the necessity of mtDNA to drive the increased basal

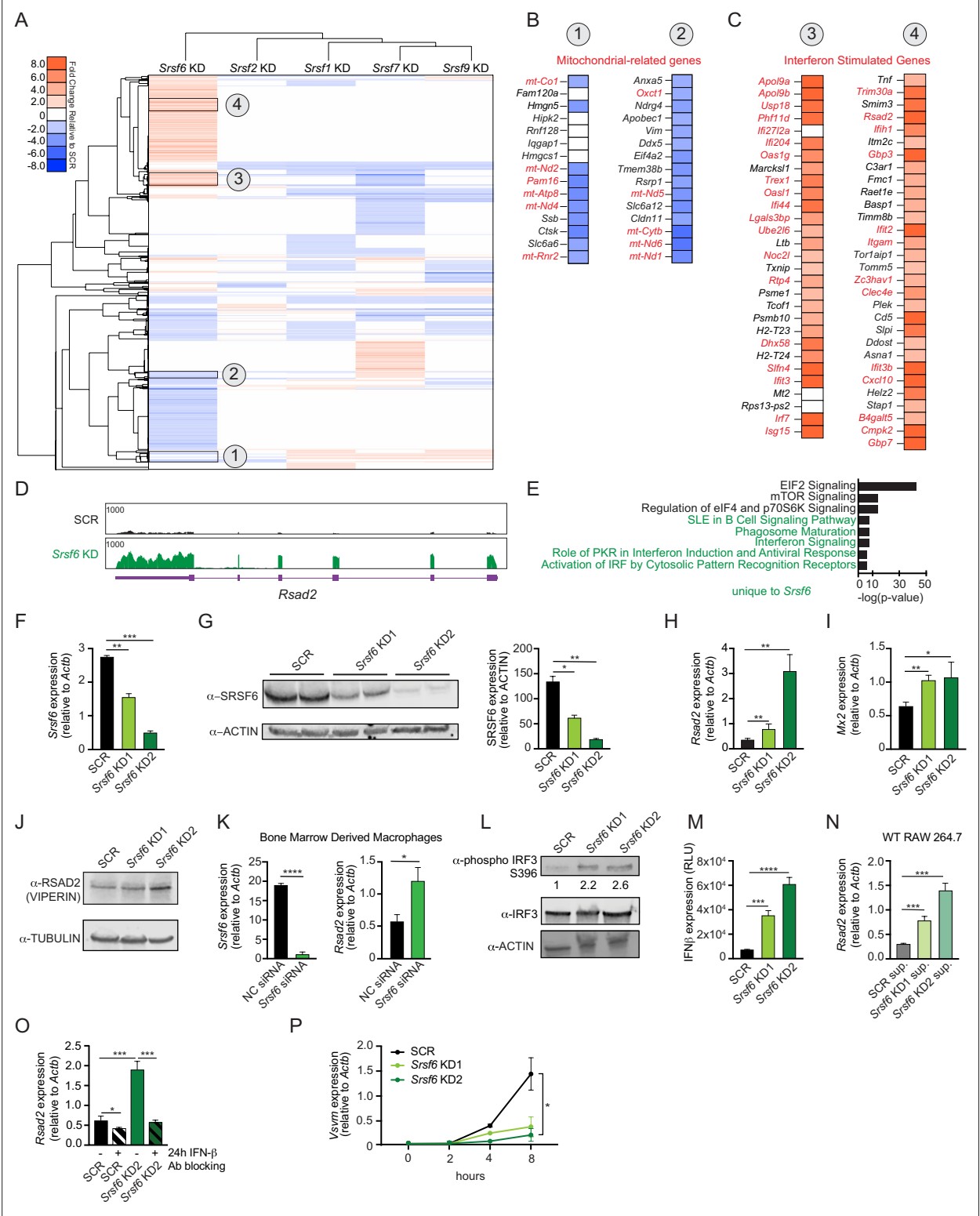

**Figure 1.** SRSF6 controls basal type I interferon expression in macrophages. (**A**) Heatmap of differentially expressed genes after knockdown of *Srsf1, 2, 6, 7,* and *9* in RAW 264.7 macrophage-like cell lines (RAW MΦ) relative to a scramble (SCR) shRNA control. (**B**) Differential gene expression of mitochondria related genes (red) in *Srsf6* KD RAW MΦ. (**C**) As in B but highlighting ISGs (red). (**D**) Integrative Genomics Viewer (IGV) tracks of *Rsad2* from *Srsf6* KD macrophage RNA seq. (**E**) Ingenuity Pathway Analysis showing canonical pathways from *Srsf6* KD RAW MΦ RNA seq. Green indicates pathways unique to SRSF6. (**F**) RT-qPCR of *Srsf6* in *Srsf6* KD RAW MΦ. (**G**) Immunoblot of SRSF6 in *Srsf6* KD RAW MΦ. (**H**) RT-qPCR of *Rsad2* in *Srsf6*

*Figure 1 continued on next page*

*Figure 1 continued*

KD RAW MΦ. (**I**) RT-qPCR of *Mx2* in *Srsf6* KD RAW MΦ. (**J**) Immunoblot of RSAD2 (VIPERIN) in *Srsf6* KD RAW MΦ. (**K**) RT-qPCR of *Srsf6* and *Rsad2* in *Srsf6* siRNA KD BMDMs compared with a negative control (NC) siRNA control. (**L**) As in G but for phosphorylated IRF3 and total IRF3. Numbers indicate densiometric measurements of pIRF3 (LICOR). (**M**) Protein quantification of extracellular IFNβ in *Srsf6* KD RAW MΦ measured by relative light units (RLU). (**N**) RT-qPCR of *Rsad2* in WT RAW MΦ incubated with SCR or *Srsf6* KD RAW MΦ supernatants for 24 h. (**O**) RT-qPCR of *Rsad2* in *Srsf6* KD RAW MΦ given IFNβ neutralizing antibody treatment for 24 h. (**P**) VSV replication in *Srsf6* KD RAW MΦ at 0, 2, 4, 8 hr post infection (MOI = 1) measured by RT-qPCR of *Vsvm*. All data are compared with a SCR control unless indicated. Data are expressed as a mean of three or more biological replicates with error bars depicting SEM. Statistical significance was determined using two tailed unpaired student's *t* test. *=p < 0.05, **=p < 0.01, ***=p < 0.001, ****=p < 0.0001.

The online version of this article includes the following source data and figure supplement(s) for figure 1:

**Source data 1.** Unmodified immunoblots of SRSF6 and ACTIN in *Srsf6* KD RAW MΦ.

**Figure supplement 1.** Loss of SRSF6 upregulates interferon stimulated genes.

ISG expression in *Srsf6* KD macrophages, as mtDNA depletion via treatment with the nucleoside analog 2,3'-dideoxycytidine (ddC) (*Figure 2C*), returned expression of the ISG *Rsad2* to that of SCR control cells (*Figure 2D*). Elevated basal ISG expression was dependent on cytosolic DNA sensing, as *Srsf6* KD did not cause *Rsad2* transcript accumulation in the absence of the cytosolic DNA sensor, cGAS (*Figure 2E–F*, *Figure 2—figure supplement 1A*). Together, these data argue that loss of SRSF6 causes leakage of mtDNA into the cytosol where it engages cGAS, leading to phosphorylation of IRF3 (*Figure 1L*), expression of IFN-β (*Figure 1M*), and activation of ISG expression in resting macrophages (*Figure 1A–C*).

To understand how loss of SRSF6 disrupts mitochondrial homeostasis to allow for mtDNA cytosolic access, we first visualized the mitochondrial network in *Srsf6* KD and control MEFs, which are well-suited for imaging due to their extensive mitochondrial network. Immunofluorescence microscopy (anti-TOM20) revealed increased mitochondrial fragmentation in *Srsf6* KD MEFs (*Figure 2G*), although total levels of mitochondrial proteins like TOM20 and VDAC were comparable between SCR and *Srsf6* KD cell lines (*Figure 2—figure supplement 1B*). Importantly, mitochondria in *Srsf6* KD RAW MΦ also exhibited loss of membrane potential (*Figure 2H*), as measured by tetramethylrhodamine ethyl ester (TMRE) signal (decreased TMRE signal = increased membrane depolarization). As another measure of mitochondrial function, we measured metabolic output in *Srsf6* KD vs. SCR RAW MΦ cells normalized to protein abundance via the Agilent Seahorse Metabolic Flex Analyzer. *Srsf6* KD cells displayed deficiencies in multiple readouts of oxygen consumption rate (OCR), which serves as a proxy for oxidative phosphorylation (OXPHOS). Loss of SRSF6 reduced mitochondrial OXPHOS, indicated by lower basal and maximal respiration. Additionally, *Srsf6* KD mitochondria had lower ATP production and lower respiratory capacity compared with SCR controls (*Figure 2I* top-J). Glycolysis, as measured by extracellular acidification rate (ECAR) was unaffected by SRSF6 knockdown (*Figure 2I*, bottom), suggesting that SRSF6 specifically impacts cellular metabolism primarily at the level of mitochondrial OXPHOS.

## SRSF6-dependent alternative splicing of BAX regulates basal ISG expression in macrophages

Our data suggest that SRSF6 limits basal type I IFN expression in macrophages through a role in maintaining mitochondrial homeostasis. SRSF6 is best known as a regulator of alternative splicing (*Filippov et al., 2008*; *Juan-Mateu et al., 2018*; *Tran and Roesser, 2003*; *Tranell et al., 2010*). Therefore, we hypothesized that loss of SRSF6 could alter alternative splicing of one or more pre-mRNAs that encode proteins involved in mitochondrial biology. To identify SRSF6-dependent alternative splicing events, we mined a list of local splicing variations (LSVs) in *Srsf6* KD MΦ cells quantified by the computational algorithm MAJIQ (Modeling Alternative Junction Inclusion Quantification) *Vaquero-Garcia et al., 2016* as reported in *Wagner et al., 2021*. As expected for an SR protein, SRSF6 mostly controls exon inclusion in macrophages (1043 exon skipping LSVs), with changes in intron retention (247 LSVs), and alternative 5' (206 LSVs) and 3' (233 LSVs) splice site usage also detected (*Figure 3A*). Manual annotation of these >1600 LSVs identified alternative splicing events in 31 genes that also demonstrated some change in gene expression (p<0.05) with annotated roles in mitochondrial function (*Figure 3B*). Semi-quantitative RT-PCR was used to validate hits and SRSF6-dependent splicing of intron 2 in the *Xaf1* pre-mRNA, which is predicted to introduce a premature stop codon and target *Xaf1* for nonsense mediated decay, was confirmed (*Figure 3—figure supplement 1A*).

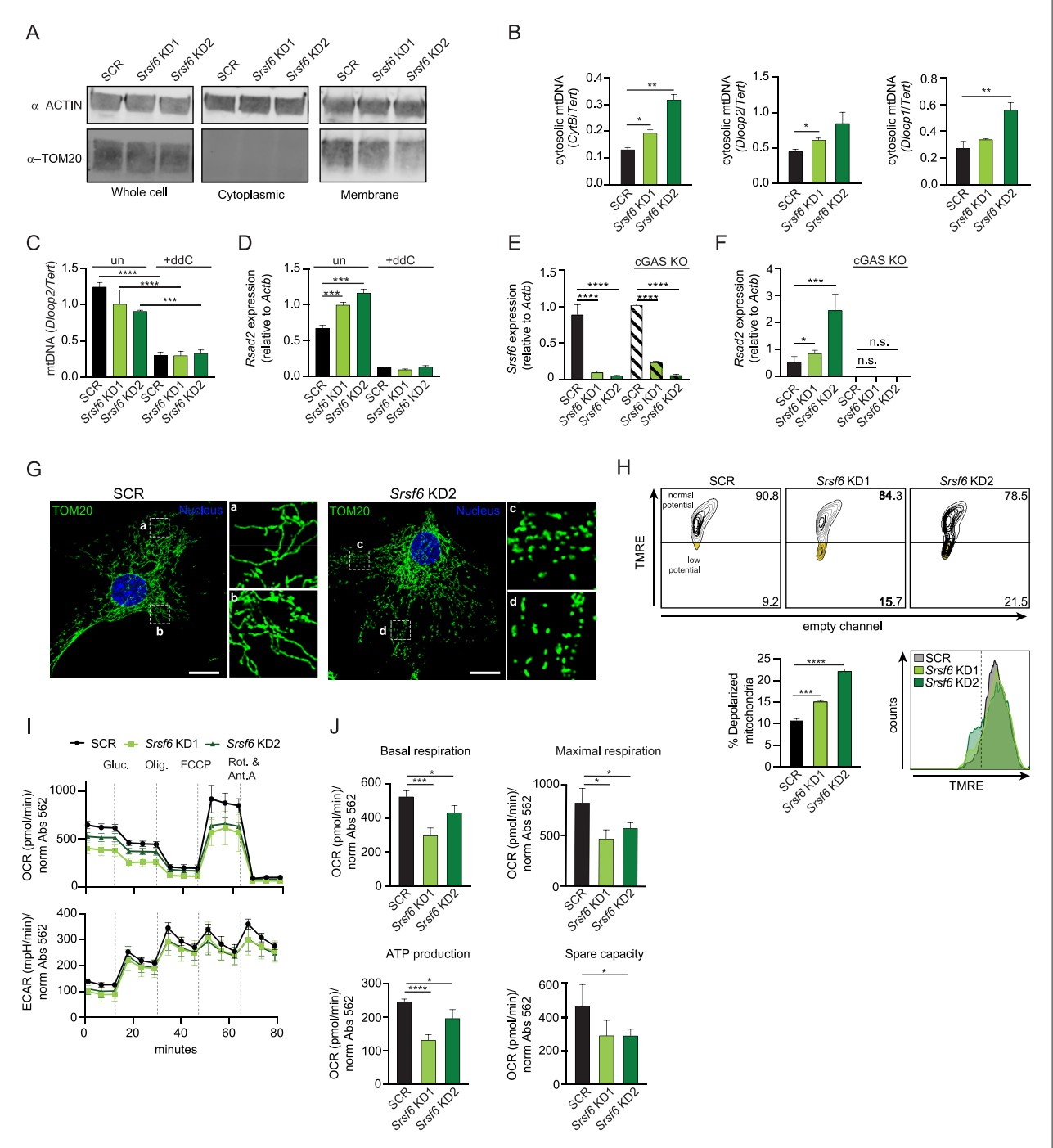

**Figure 2.** SRSF6 limits cytosolic mtDNA release by maintaining mitochondrial homeostasis. (**A**) Immunoblot of mitochondria (TOM20) in total, cytoplasmic, and membrane fractions of *Srsf6* KD RAW MΦ. (**B**) RT-qPCR of mtDNAs *CytB, Dloop1, Dloop2* relative to nuclear DNA *Tert* in cytosolic fractions of *Srsf6* KD RAW MΦ. (**C**) RT-qPCR of total mtDNA *Dloop2* relative to nuclear DNA *Tert* in *Srsf6* KD and SCR control RAW MΦ with or without mtDNA depletion for 8 days. (**D**) As in C but measuring *Rsad2*. (**E**) RT-qPCR of *Srsf6* in cGAS KO RAW MΦ. (**F**) As in E but measuring *Rsad2*. (**G**) Immunofluorescence microscopy images visualizing mitochondria in *Srsf6* KD MEFs immunostained with TOM20. Scale bar = 10 μm. (**H**) Mitochondria membrane potential measured by TMRE staining of *Srsf6* KD RAW MΦ. (**I**) Oxygen consumption rate (OCAR) and Extracellular acidification rate (ECAR) measured by Seahorse in *Srsf6* KD RAW MΦ. (**J**) Basal respiration, maximal respiration, ATP production, and spare capacity of *Srsf6* KD RAW MΦ determined by OCAR analysis. All data are compared with a SCR control unless indicated. Data are expressed as a mean of three or more biological replicates with error bars depicting SEM. Statistical significance was determined using two tailed unpaired student's *t* test. *=p < 0.05, **=p < 0.01, ***=p < 0.001, ****=p < 0.0001.

*Figure 2 continued on next page*

*Figure 2 continued*

The online version of this article includes the following source data and figure supplement(s) for figure 2:

**Source data 1.** Unmodified immunoblots of ACTIN and mitochondria (TOM20) in total, cytoplasmic, and membrane fractions of *Srsf6* KD RAW MΦ.

**Figure supplement 1.** Mitochondrial protein levels in *Srsf6* KD macrophages.

**Figure supplement 1—source data 1.** Unmodified immunoblots of ACTIN and cGAS in WT and *cGAS* KO RAW MΦ.

**Figure supplement 1—source data 2.** Unmodified immunoblots of VDAC1 and TOM20 in *Srsf6* KD RAW MΦ.

One alternative splicing event with unique protein coding capacity that piqued our interest occurred in *Bax*, the mitochondrial pore-forming protein and executioner of apoptosis. Specifically, MAJIQ analysis measured preferential retention of intron 1 in *Srsf6* KD macrophages (**Figure 3C**). Considerable *Bax* intron 1 retention occurred in SCR MΦ cells as well, but to a lesser extent (**Figure 3C–E**). We validated SRSF6-dependent *Bax* splicing by RT-qPCR, using a forward primer in exon 1 and a reverse primer in intron 1 (**Figure 3F**), as well as by semi-quantitative RT-PCR, using a forward primer in exon 1 and a reverse primer in exon 6 (**Figure 3—figure supplement 1B**). Increased *Bax* alternative splicing did not impact total BAX protein expression (**Figure 3—figure supplement 1C**). Retention of intron 1 in *Bax* causes usage of a downstream ATG start site encoded in exon 2, which creates a 20 amino acid N-terminal truncation, but otherwise leaves the *Bax* coding sequence completely in-frame and intact (**Figure 3G** and **Figure 3—figure supplement 1D**). This annotated isoform of BAX, dubbed BAX kappa (BAX-κ; GenBank accession number AY095934), is upregulated in the rat hippocampus following cerebral ischemic injury and promotes apoptosis when overexpressed in murine hippocampal neuronal cells (**Jin et al., 2001**). Because they both lack an ART (apoptosis-regulating targeting) sequence (**Cartron et al., 2005**; **Goping et al., 1998**), Bax-κ is predicted to be functionally analogous to the human BAX-psi isoform, which has been shown to constitutively associate with mitochondria (**Cartron et al., 2005**). ESE Finder, a web-based platform to identify exonic splicing enhancer motifs (**Cartegni et al., 2003**), identified three strong SRSF6 consensus sequences in *Bax* exons 1, 3, and 6 (**Figure 3H**). UV Crosslinking-immunoprecipitation (CLIP) experiments in RAW MΦ cells showed a clear enrichment of *Bax* transcripts bound to 3xFLAG-SRSF6 compared with a 3xFLAG-GFP control (**Figure 3I–J**). Together, these data strongly suggest that SRSF6 controls splicing of *Bax* intron 1 by directly binding consensus exonic splicing enhancers in *Bax* exons.

Previous literature has demonstrated that in addition to promoting release of cytochrome c during apoptosis, BAX pores can release mtDNA capable of stimulating cGAS-dependent type I IFN responses (**Rongvaux, 2018**; **Rongvaux et al., 2014**). We hypothesized that high basal ISGs in *Srsf6* KD macrophages are a consequence of Bax-κ-dependent release of mtDNA. Consistent with this prediction, even modest knockdown of *Bax* (all isoforms) via siRNA transfection rescued high *Rsad2* and *Ifit1* expression in *Srsf6* KD macrophages (**Figure 3K**). Because knockdown of *Bax* in wild-type RAW MΦ had no impact on *Rsad2* or *Isg15* expression (**Figure 3L**), we can conclude that links between aberrant basal ISG expression and BAX are limited to *Srsf6* KD cells in which *Bax* splicing is defective.

## SRSF6 deficiency sensitizes macrophages to cell death

In addition to their role in cellular metabolism and energy production, mitochondria serve as gatekeepers of multiple cell death pathways. Based on previous studies of BAX-κ and human BAX-psi, we hypothesized that accumulation of BAX-κ would promote cell death in RAW MΦ cells. We first tested this by measuring incorporation of the cell viability stain propidium iodide (PI) in resting SCR and Srsf6 KD MΦ cells. We observed significantly higher levels of PI incorporation in *Srsf6* KD cell lines (**Figure 4A**), suggesting that even in the absence of cell death inducing agents, a population of stable *Srsf6* KD MΦ undergoes cell death. Because *Srsf6* KD MΦ secretes unusually high levels of IFN-β, we tested whether cell death was dependent on IFN-β as in **Apelbaum et al., 2013**; **Sarhan et al., 2019**. Treatment with an anti-IFN-β neutralizing antibody had no effect on PI incorporation in SCR or *Srsf6* KD cells, effectively ruling out a role for IFN−β in mediating SRSF6-dependent cell death (**Figure 4B**). Flow cytometric analysis of *Srsf6* KD cells via PI and annexin V staining (a protein that preferentially binds to phosphatidylserine on cells undergoing apoptosis), confirmed higher levels of PI incorporation in the absence of *Srsf6* (**Figure 4C–D**). It also identified a population of pro-apoptotic cells (annexin V-positive but PI-negative) that is more abundant in *Srsf6* KDs (6.3% in KD1 and 9.7% in

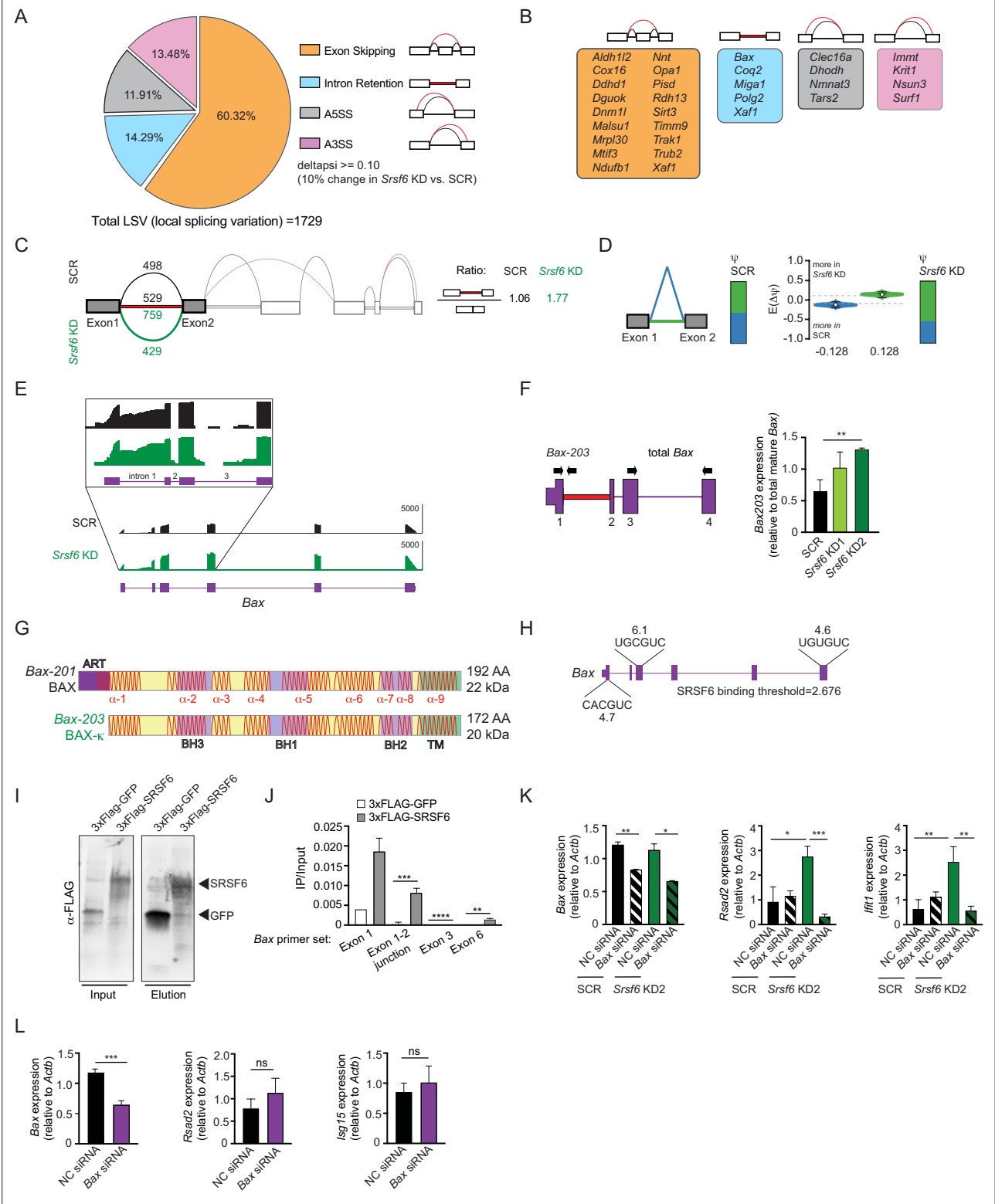

**Figure 3.** SRSF6 controls alternative splicing of the mitochondrial apoptotic factor BAX. (**A**) Percentages of alternative splicing (AS) events in *Srsf6* KD RAW MΦ (deltapsi ≥ 0.1). (**B**) Categorization of alternative splicing events in mitochondria genes differentially expressed in *Srsf6* KD RAW MΦ. Red lines are AS events and black lines are WT events. (**C**) Splice graph of *Bax* in SCR (top) and *Srsf6* KD (bottom) RAW MΦ generated by MAJIQ/VOILA. Intron 1 retention reads relative to exon1-2 junction reads in each genotype shown on right. (**D**) MAJIQ Ψ quantification of junctions as illustrated in (**C**) from SCR (left) and *Srsf6* KD (right) RAW MΦ. Intron retention displayed in green; intron removal displayed in blue. (**E**) Integrative Genomics Viewer (IGV)

*Figure 3 continued on next page*

*Figure 3 continued*

tracks of *Bax,* highlighting exon 1 to exon 3. Zoom-in (top) uses a log scale to facilitate appreciation of the intron reads. (**F**) RT-qPCR of *Bax*203 relative to mature *Bax* expression in *Srsf6* KD RAW MΦ. Primers shown on schematic. (**G**) Schematics of BAX and BAX-κ proteins. Alpha-helical domains shown as red lines. ART = apoptosis regulatory targeting domain (***Goping et al., 1998***). (**H**) Diagram of predicted *Srsf6* binding sites in *Bax* pre-mRNA with predicted binding strength scores (from ESE Finder). (**I**) CLIP Immunoblot of 3xFLAG-GFP and 3xFLAG-SRSF6 constructs expressed in RAW MΦ for 24 h. (**J**) CLIP RT-qPCR of 3xFLAG-GFP and 3xFLAG-SRSF6 RT-qPCR of *Bax* exon 1, exon1-2 junction, exon 3, and exon 6. Data shown as IP relative to input. (**K**) RT-qPCR of *Bax*, *Rsad2,* and *Isg15* in *Srsf6* KD RAW MΦ with *Bax* KD via siRNA transfection. (**L**) RT-qPCR of *Bax, Rsad2,* and *Isg15* in transient *Bax* KD RAW MΦ. All data are compared with a SCR control unless indicated. Data are expressed as a mean of three or more biological replicates with error bars depicting SEM. Statistical significance was determined using two tailed unpaired student's *t* test. *=p < 0.05, **=p < 0.01, ***=p < 0.001, ****=p < 0.0001.

The online version of this article includes the following source data and figure supplement(s) for figure 3:

**Source data 1.** Unmodified immunoblot of FLAG of 3xFLAG-GFP and 3xFLAG-SRSF6 constructs expressed in RAW MΦ.

**Figure supplement 1.** Loss of SRSF6 impacts alternative splicing of transcripts with known roles in mitochondrial biology.

**Figure supplement 1—source data 1.** Unmodified semi-quantitative RT-PCR gel of *Xaf1* in *Srsf6* KD RAW MΦ.

**Figure supplement 1—source data 2.** Unmodified semi-quantitative RT-PCR gels of *Bax* and *Brd2* (control) in *Srsf6* KD RAW MΦ.

**Figure supplement 1—source data 3.** Unmodified immunoblot of BAX in *Srsf6* KD RAW MΦ.

KD2 vs. 4.2% in SCR) (***Figure 4E***). Consistent with *Srsf6* KD RAW 264.7 cells being prone to apoptosis, significantly more *Srsf6* cells stained PI +after treatment with low levels of the apoptosis-inducing drugs staurosporine (***Figure 4F***) or ABT373 (***Figure 4G***) and at early time-points following treatment with the cell death agonist etoposide (***Figure 4—figure supplement 1A***).

We next set out to better define the nature of cell death in *Srsf6* RAW MΦ cells. Previous reports have implicated BAX in non-canonical proinflammatory forms of cell death and IL-1β release mediated by caspase 8 (***Hu et al., 2020***; ***Vince et al., 2018***). We did not suspect inflammasome-mediated cell death could explain our findings in *Srsf6* KD RAW MΦ cells as RAW cells lack the inflammasome adapter ASC (***Pelegrin et al., 2008***). However, given that the cell death observed was not silent, with ISG upregulation, we sought to rule out additional proinflammatory pathways present during *Srsf6*-mediated cell death. We detected no significant difference in IL-1β release between BMDMs transfected with a non-targeting siRNA vs. *Srsf6*-targeting siRNA in either unprimed or poly dA:dT primed cells (***Figure 4—figure supplement 1B-C***). Curiously, despite displaying an apoptotic signature (***Figure 4C, E***), cell death in *Srsf6* KD RAW MΦ cells was caspase-independent, as treatment with the pan-caspase inhibitors Q-VD-OPh (***Figure 4H***) and Z-VAD-FMK (***Figure 4—figure supplement 1D***) had no impact on PI incorporation in resting *Srsf6* KD cells. Consistent with lack of caspase involvement, resting *Srsf6* KD RAW MΦ did not release cytochrome c (***Figure 4I***). These observations, coupled with the disruption to mitochondrial membrane potential reported in (***Figure 2H***), suggest a form of mitochondrial-dependent caspase-independent cell death (CICD) (as described by ***Xiang et al., 1996***, reviewed in ***Tait and Green, 2008***) that occurs preferentially in *Srsf6 KD* macrophages. We hypothesize that BAX-κ, which can trigger mitochondrial outer membrane permeabilization (MOMP) and release of mtDNA without releasing cytochrome c, is a major contributor to this noncanonical form of cell death, which is characterized by low-level sustained damage to mitochondria.

We next sought to more directly associate BAX and BAX-κ with the altered mitochondrial homeostasis observed in *Srsf6* KD MΦ. Our data, and that from previous studies of *Bax-psi* (***Cartron et al., 2005***), suggest that BAX-κ is a constitutively active form of BAX that readily localizes to mitochondrial membranes in resting cells. To determine whether BAX protein accumulates on mitochondria in resting *Srsf6* KD MΦ, we performed a 'mitoFLOW' experiment as described in ***Weindel et al., 2022***. Briefly, mitoTRACKER green stained mitochondria were isolated by passive lysis and centrifugation, stained with a BAX antibody, and analyzed by flow cytometry. MitoTRACKER was used to identify and gate mitochondria and then BAX association was quantified (***Figure 4J*** top). We observed a significant increase of BAX protein on mitochondria isolated from resting *Srsf6* KD cells compared with SCR controls (***Figure 4J*** bottom). Although our antibody cannot distinguish between BAX and BAX-κ, this data further connects loss of SRSF6 with increased BAX mitochondrial association, which we argue is mainly contributed by the BAX-κ isoform (full-length BAX is predominantly cytosolic in resting cells) (***Cosentino and García-Sáez, 2017***).

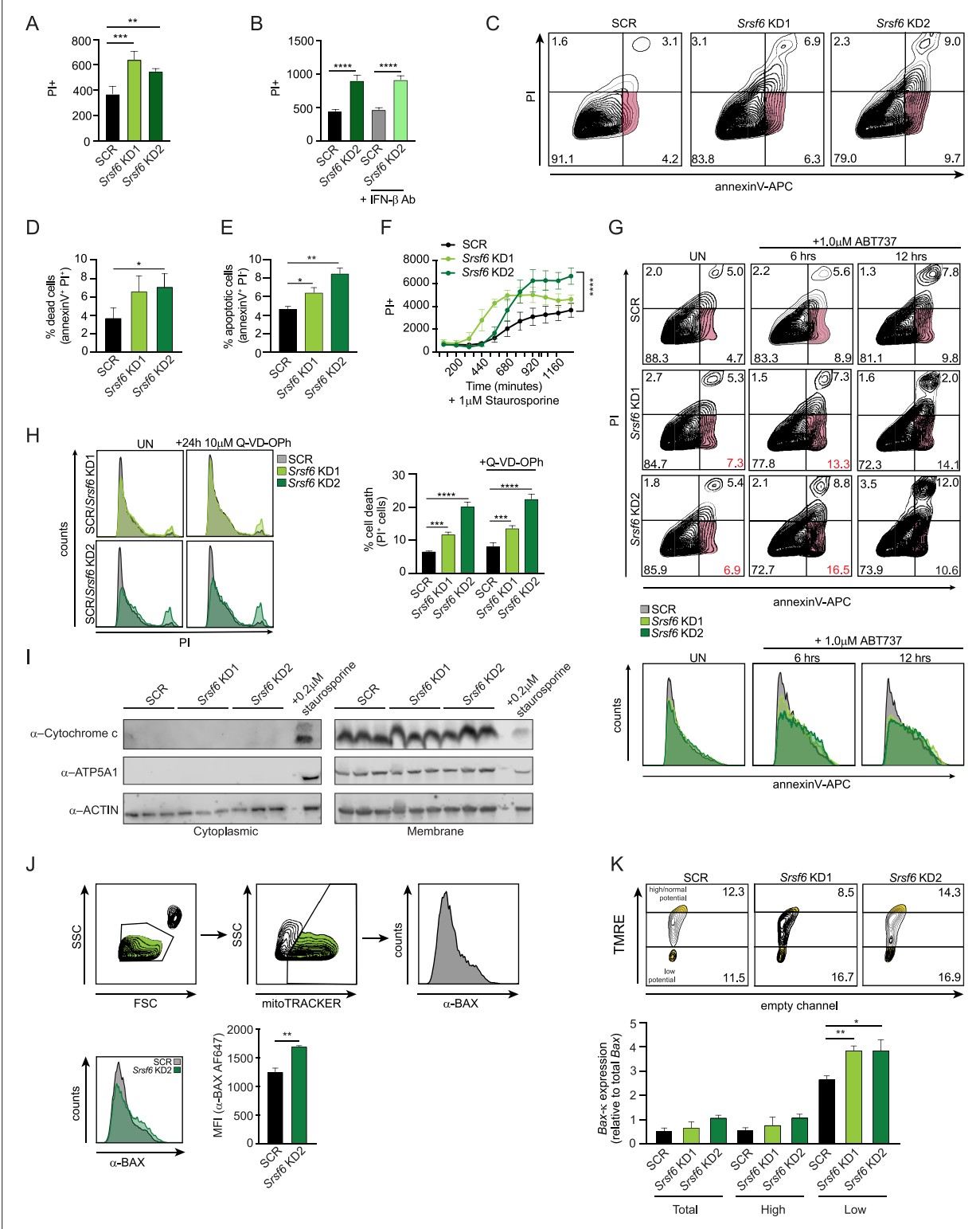

**Figure 4.** Loss of SRSF6 sensitizes macrophages to caspase-independent apoptotic cell death. (**A**) Cell death in *Srsf6* KD RAW MΦ measured by live cell imaging of propidium iodide (PI) staining. (**B**) Cell death in *Srsf6* KD RAW MΦ treated with IFN-β neutralizing antibody. (**C**) Apoptotic cell death measured by flow cytometry using APC conjugated annexin V (annexinV-APC) and propidium iodide (PI) dyes in *Srsf6* KD RAW MΦ. (**D**) Quantification of dead cells in *Srsf6* KD RAW MΦ from C. (**E**) Quantification of apoptotic cells in *Srsf6* KD RAW MΦ from C. (**F**) Cell death over a time course in *Srsf6* KD RAW 264.7 cells treated with 1 μM staurosporine. (**G**) Apoptotic cell death over a time course measured by flow cytometry using annexinV-APC and PI in *Srsf6* KD RAW MΦ treated with 1 μM ABT737. Histograms display annexinV-APC single stain in *Srsf6* KD. Red numbers indicate annexinV+/

*Figure 4 continued on next page*

*Figure 4 continued*

PI- cells in *Srsf6* KDs. (**H**) Histogram showing cell death after caspase inhibition by flow cytometry in *Srsf6* KD RAW MΦ. Cell death quantification (right). (**I**) Immunoblot of cytochrome c in cytoplasmic and membrane fractions of *Srsf6* KD RAW 264.7 cells. SCR cells treated with 0.2 μM staurosporine for 24 h used as a positive control. (**J**) Schematic of mitoFLOW workflow (top). Histogram showing BAX accumulation on *Srsf6* KD RAW MΦ isolated mitochondria (bottom) (**K**) Mitochondria membrane potential measured by TMRE staining of *Srsf6* KD RAW MΦ (top). RT-qPCR of BAX-κ relative to mature *Bax* expression in total, high, and low mitochondria membrane potential cell populations. All data are compared with a SCR control unless indicated. Data are expressed as a mean of three or more biological replicates with error bars depicting SEM. Statistical significance was determined using two tailed unpaired student's *t* test. \*=p < 0.05, \*\*=p < 0.01, \*\*\*=p < 0.001, \*\*\*\*=p < 0.0001.

The online version of this article includes the following source data and figure supplement(s) for figure 4:

**Source data 1.** Unmodified immunoblots of ATP5A1, ACTIN, and cytochrome c in cytoplasmic and membrane fractions of *Srsf6* KD RAW MΦ.

**Figure supplement 1.** *Srsf6* KD cells are sensitive to cell death agonists.

We next wanted to see if *Bax-κ* transcript expression correlated with loss of mitochondrial membrane polarization (*Figure 2H*). To this end, we sorted TMRE-treated cells into high TMRE signal (normal mitochondrial membrane potential) or low TMRE signal (low mitochondrial membrane potential/depolarized mitochondria). RNA was isolated from these two pools of cells, alongside a total, unsorted pool, and RT-qPCR was performed as in *Figure 3F* to measure *Bax-κ*. *Bax-κ* transcripts were considerably higher in *Srsf6* KD cells with low membrane potential compared with mitochondria with high membrane potential or total mitochondria (*Figure 4K*). *Bax-κ* transcripts were most abundant in low membrane potential mitochondria isolated from *Srsf6* KD cells, although notably, *Bax-κ* expression correlated with mitochondrial depolarization in SCR cells as well.

## BAX-κ is sufficient to drive basal ISG expression and apoptosis in macrophages

The results of our *Bax* KD experiment suggest that BAX is necessary to drive type I IFN expression in *Srsf6* KD cells (*Figure 3K*). To specifically implicate BAX-κ in this phenotype, we generated C-terminal 2xSTREP-tagged constructs of BAX, BAX-κ, and a BAX point mutant (BAX^G179P) that is unable to translocate to mitochondria (*Kuwana et al., 2020*) and introduced them via lentiviral transduction into RAW MΦ cells expressing a doxycycline-inducible transactivator (*Figure 5—figure supplement 1A*). Because high levels of doxycycline can negatively impact mitochondrial function (*Dijk et al., 2020*), we experimentally determined the lowest doxycycline concentration that ensured robust and uniform expression of an inducible mCherry construct without significant toxicity (1 μg/mL) (*Figure 5—figure supplement 1A-C*). All three of our BAX constructs were expressed in a doxycycline-inducible fashion, albeit to much lower levels than that of a 2xSTREP-GFP control (*Figure 5A*). We found that expression of BAX-κ was sufficient to induce robust basal ISG expression in the wild-type RAW doxycycline-inducible cell line (*Figure 5B*). Importantly, this ISG induction was dose-dependent and seen even at early timepoints post-DOX treatment when BAX-κ ectopic expression is still quite low (*Figure 5A*). BAX-κ expression was also sufficient to induce apoptosis and cell death, as measured by flow cytometry of PI and annexin V-stained cells at 15 h post-doxycycline treatment (*Figure 5C*) or by PI + incorporation over a 15 hr time-course of doxycycline induction (*Figure 5D*). The accumulation of PI + and annexin V + cells was dependent on the dose of BAX-κ. Unlike the cell death we measured in resting *Srsf6* KD RAW MΦ cells, BAX-κ induced apoptosis and cell death at 15 hr (*Figure 5C*) or 24 hr (*Figure 5—figure supplement 1D*) post-DOX treatment was largely caspase-dependent (*Figure 5C*, bottom graphs). We suspect this contrast between *Srsf6* KD cells and BAX-κ overexpressing cells is due in large part to the timing and dose of BAX-κ: caspase-inhibition had only a slight impact at 5 hr post-DOX, with a reduction of annexin V+/PI- cells from 8.42% to 6.94% (when BAX-κ expression is low), compared with a rescue of annexin V+/PI- cells from 30% to 12.1% at 24 hr (when BAX-κexpression is high) (*Figure 5—figure supplement 1D*). We predict that high levels of BAX-κ in these overexpression cell lines at later time points induces a level of MOMP that is sufficient to release cytochrome c and activate downstream caspases. When we directly stimulated apoptosis in these cells via low levels of staurosporine (1 μM), BAX and BAX-κ expression induced similar amounts of enhanced cell death relative to GFP or BAX^G179P-expressing cells, confirming that an activation signal is required for full length BAX to stimulate cell death (*Figure 5E*).

Last, we sought to measure the amount of total BAX associated with mitochondria in each of the overexpression cell lines. Using mitoFLOW (as in *Figure 4J*), we measured significantly more

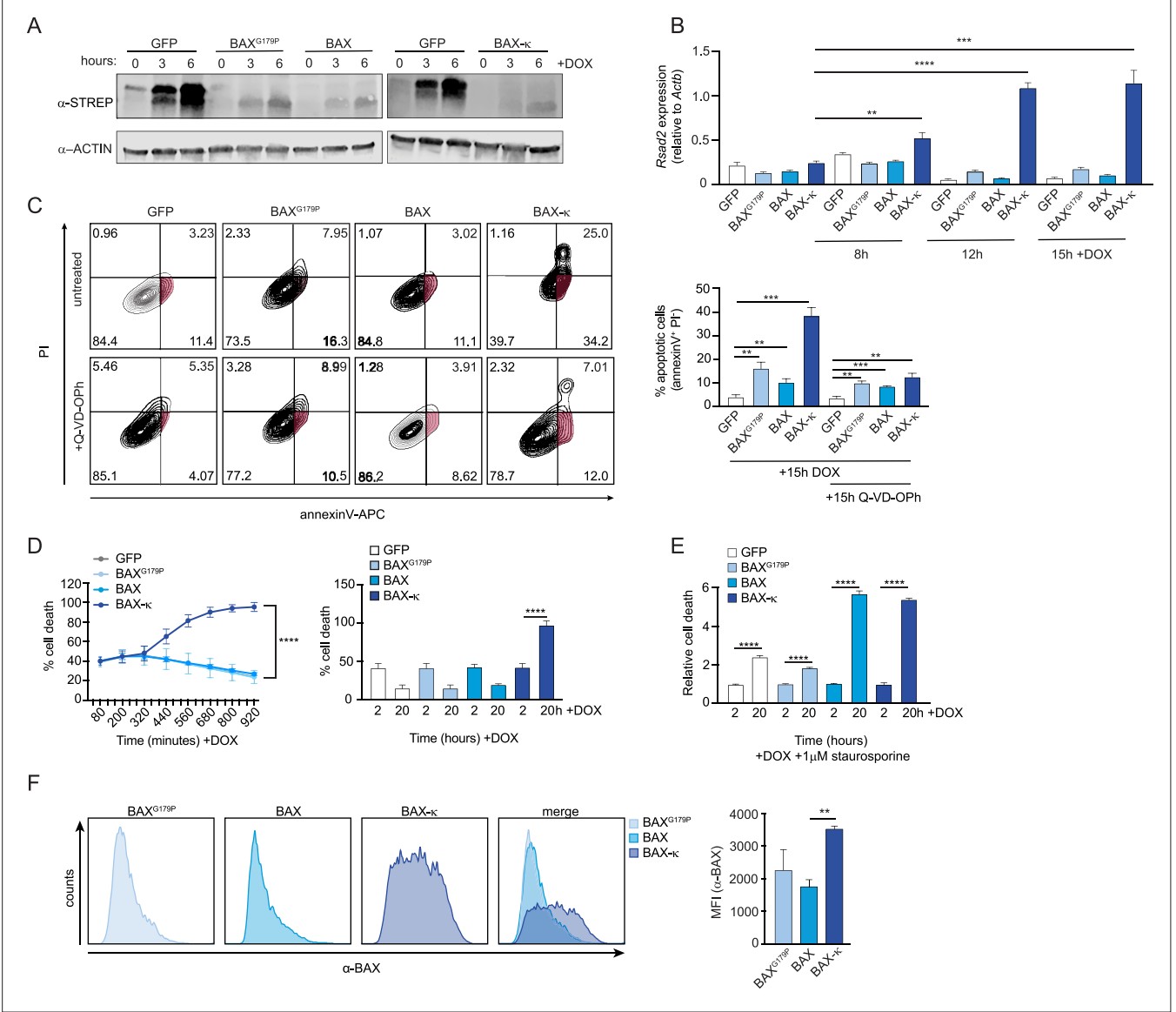

**Figure 5.** Expression of BAX-κ promotes type I IFN expression and cell death in macrophages. (**A**) Immunoblot of strep tagged BAX[G179P], BAX, and BAX-κ inducible RAW MΦ expressed over a time course after addition of doxycycline (DOX). (**B**) Expression of *Rsad2* over a time course of 8, 12, and 15 hr after DOX induction in GFP, BAX[G179P], BAX, and BAX-κ-expressing RAW MΦ by RT-qPCR. (**C**) Apoptotic cell death measured by flow cytometry using annexinV-APC and PI in GFP, BAX[G179P], BAX, and BAX-κ inducible macrophages expressed for 15 hr with 1 µg DOX and caspase inhibitor (10 µM Q-VD-OPh). Apoptotic cells (AnnexinV+/PI-) quantification (right). (**D**) Cell death over a time course after DOX induced expression of GFP, BAX[G179P], BAX, and BAX-κ. Starting and ending cell death (PI+) shown as a bar graph on right. (**E**) Relative cell death measured by PI incorporation at 2 and 20 hr after DOX-induced expression of GFP, BAX[G179P], BAX, and BAX + addition of 1 µM staurosporine. (**F**) Histogram showing BAX accumulation on 20 h DOX-induced GFP, BAX[G179P], BAX, and BAX isolated mitochondria. Data are expressed as a mean of three or more biological replicates with error bars depicting SEM. Statistical significance was determined using two tailed unpaired student's *t* test. *=p < 0.05, **=p < 0.01, ***=p < 0.001, ****=p < 0.0001.

The online version of this article includes the following source data and figure supplement(s) for figure 5:

**Source data 1.** Unmodified immunoblot of ACTIN and STREP tagged BAX[G179P], BAX, and BAX-κ inducible RAW MΦ expressed over a time course after addition of doxycycline (DOX).

**Figure supplement 1.** Bax-κ expression induces cell death.

mitochondria-associated BAX in BAX-κ expressing cells compared with cells expressing BAX[G179P] or BAX (*Figure 5F*). Collectively, these data argue that expression of BAX-κ is sufficient to drive ISG expression and cell death in resting macrophages and support a major role for BAX-κ in inducing the mitochondrial phenotypes we uncovered in *Srsf6* KD macrophages.

## SRSF6 phosphorylation regulates *Bax* alternative splicing and cell death

Our original interest in SR protein function in macrophages was borne out of a global phosphoproteomics analysis that reported SR proteins, including SRSF6, were differentially phosphorylated in BMDMs at several sites over a 24 hr time course of infection with the intracellular bacterial pathogen *Mycobacterium tuberculosis* (Mtb) (*Budzik et al., 2020*). Having discovered a role for SRSF6 in

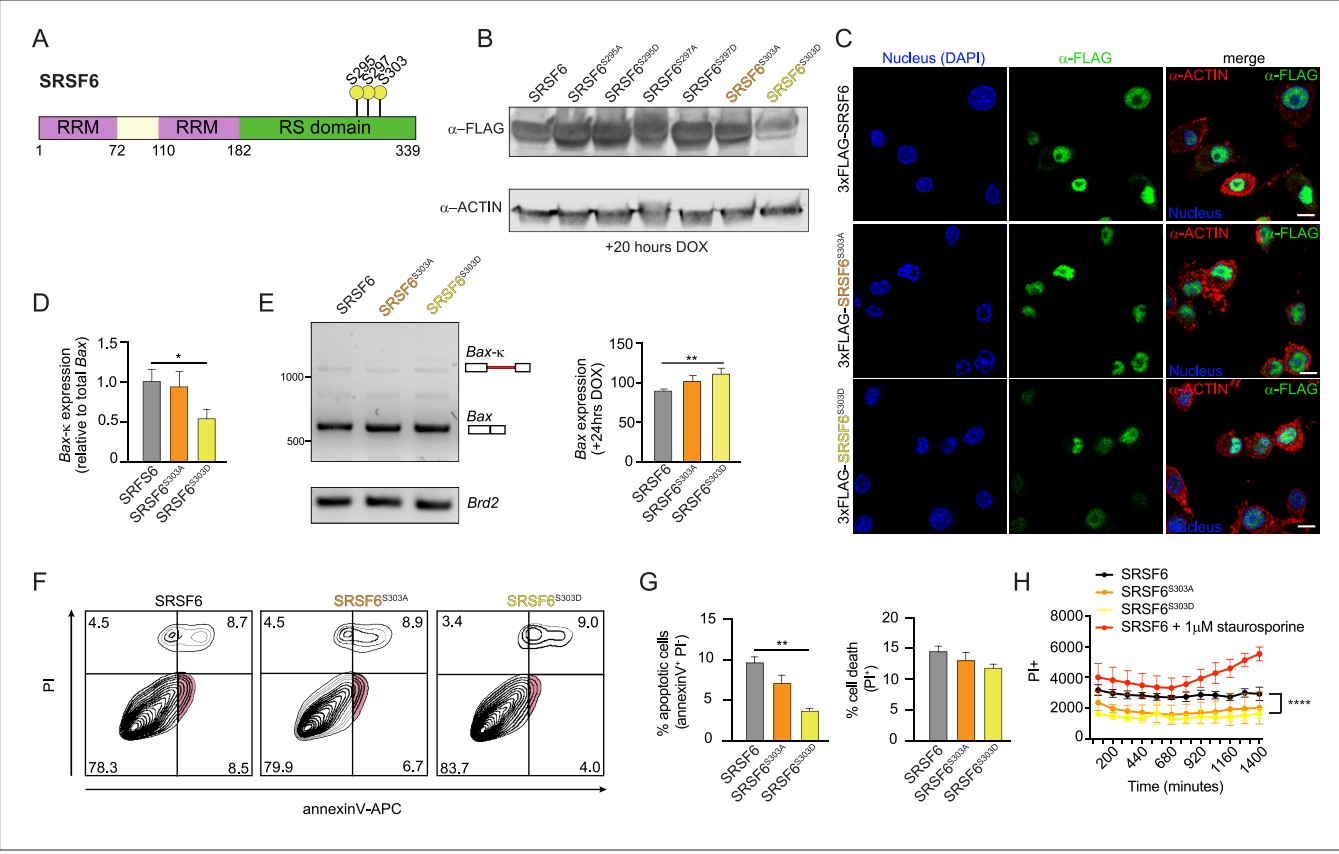

**Figure 6.** Phosphorylation of SRSF6 at S303 promotes splicing of *Bax* to limit *Bax-κ* expression and prevent cell death. (**A**) Diagram of differentially phosphorylated residues in SRSF6 according to *Budzik et al., 2020*. (**B**) Immunoblot of FLAG tagged SRSF6, SRSF6[S295A], SRSF6[295D], SRSF6[S297A], SRSF6[S297D], SRSF6[S303A], and SRSF6[S303D] inducible RAW MΦ expressed for 24 hr after DOX induction. (**C**) Immunofluorescence microscopy images visualizing 3x-FLAG tagged SRSF6, SRSF6[S303A], and SRSF6[S303D] inducible RAW MΦ expressed for 24 hr after DOX induction. Scale bar = 10 μm. (**D**) RT-qPCR of *Bax*203 in FLAG-tagged SRSF6, SRSF6[S303A], and SRSF6[S303D] inducible RAW MΦ after DOX induction for 24 hr. (**E**) Semi-quantitative RT-PCR of *Bax* and *Brd2* (control) in FLAG-tagged SRSF6, SRSF6[S303A], and SRSF6[S303D] inducible RAW MΦ expressed for 24 hr after DOX induction with quantification of multiple independent experiment. Representative gel shown. (**F**) Apoptotic cell death measured by flow cytometry using annexinV-APC and PI in FLAG tagged SRSF6, SRSF6[S303A], and SRSF6[S303D] inducible RAW MΦ expressed for 24 hr after DOX induction. (**G**) % apoptotic cells and % dead cells from D. (**H**) Cell death over a time course in FLAG tagged SRSF6, SRSF6[S303A], and SRSF6[S303D] inducible RAW MΦ. FLAG-tagged SRSF6 inducible RAW MΦ were treated with 1 μM staurosporine as a positive control. Data are expressed as a mean of three or more biological replicates with error bars depicting SEM. Statistical significance was determined using two tailed unpaired student's *t* test. *=p < 0.05, **=p < 0.01, ***=p < 0.001, ****=p < 0.0001.

The online version of this article includes the following source data and figure supplement(s) for figure 6:

**Source data 1.** Unmodified immunoblot of ACTIN and FLAG tagged SRSF6, SRSF6[S295A], SRSF6[295D], SRSF6[S297A], SRSF6[S297D], SRSF6[S303A], and SRSF6[S303D] inducible RAW MΦ expressed for 24 h after DOX induction.

**Figure supplement 1.** Expression of SRSF6-S303D reduces Bax-κ expression.

alternative splicing of BAX and regulating programmed cell death, we set out to determine whether phosphorylation of SRSF6 impacted its ability to carry out these activities in macrophages. Leveraging data from *Budzik et al., 2020*, we prioritized three serine residues that were differentially phosphorylated in studies of Mtb-infected or LPS-stimulated macrophages *Weintz et al., 2010*: S295, S297, and S303 (*Figure 6A*). To test the contribution of phosphorylation of these serines to SRSF6 function, we generated RAW MΦ cell lines expressing doxycycline-inducible constructs of wild-type SRSF6 alongside phosphodead (Ser(S)-to-Ala(A)) or phosphomimetic (Ser(S)-to-Asp(D)) mutations at each site. Expression of each SRSF6 allele was confirmed by immunoblot at 24 hr following addition of doxycycline (*Figure 6B*). Because phosphorylation of SRSF6 has been implicated in nuclear/cytoplasmic shuttling (*Jeong, 2017*), we tested where each SRSF6 mutant accumulated in cells using confocal immunofluorescence microscopy, using an antibody directed against the 3xFLAG tag. We observed no major differences in SRSF6 protein localization in any of our phosphomutant/mimetic cell lines compared with wild-type SRSF6 (predominantly nuclear in all cases) (*Figure 6C*).

To begin to implicate specific phosphorylation events in SRSF6 activity, we measured the ratio of *Bax*-κ relative to total *Bax*, as in *Figure 3G*, in each of the SRSF6-expressing cell lines. We observed that expression of SRSF6^S303D decreased levels of *Bax*-κ transcripts, compared with those in wild-type SRSF6-expressing macrophages (*Figure 6—figure supplement 1A* and *Figure 6D*). Likewise, SRSF6^S303D-expressing cell lines had higher levels of canonical *Bax* transcripts, relative to wild-type SRSF6-expressing cells, by semi-quantitative RT-PCR (*Figure 6E*). Consistent with altered Bax-κ to Bax ratios in SRSF6^S303D-expressing cell lines, expression of the SRSF6^S303D phosphomimetic allele rendered cells less prone to apoptosis (annexin V+/PI-) and cell death (*Figure 6F–H*). Together, these data

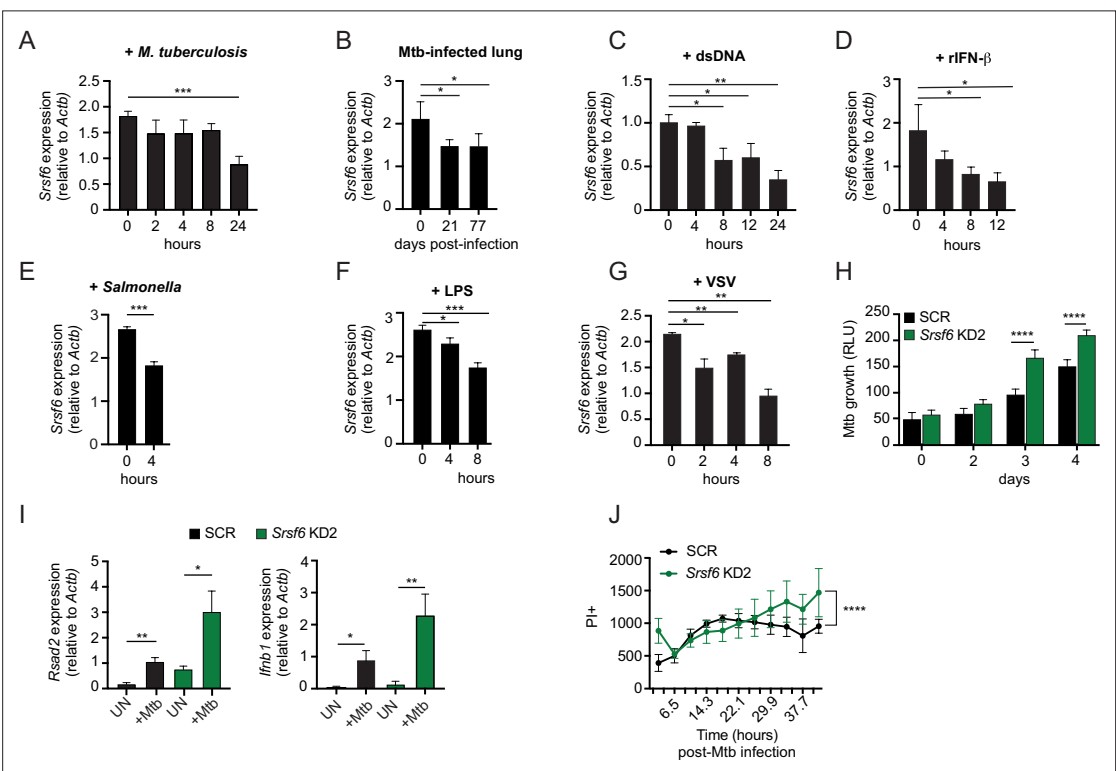

**Figure 7.** Modulation of SRSF6 expression contributes to innate immune control of the intracellular bacterial pathogen *M. tuberculosis*. (**A**) RT-qPCR of *Srsf6* in RAW MΦ infected with *M. tuberculosis* (Mtb) (MOI = 5) over a time course. (**B**) RT-qPCR of *Srsf6* in Mtb-infected mouse lung samples over a time course of *in vivo* infection. (**C**) RT-qPCR of *Srsf6* in RAW MΦ treated with 1 μg double stranded DNA (dsDNA). over a time course. (**D**) As in C but treated with recombinant IFN-β (rIFN-β). (**E**) RT-qPCR of *Srsf6* in *S. enterica* (Typhimurium) infected RAW MΦ (MOI = 5) at 0 and 4 hr. (**F**) As in C but treated with LPS. (**G**) RT-qPCR of *Srsf6* in VSV infected RAW MΦ (MOI = 1) over a time course. (**H**) Mtb *luxBCADE* growth in *Srsf6* KD RAW MΦ measured by relative light units (RLUs) over a time course (MOI = 1). (**I**) RT-qPCR of *Rsad2* and *Ifnb1* in *Srsf6* KD RAW MΦ infected with Mtb at (MOI = 10), 4 hr post-infection. (**J**) Cell death over a time course in SCR and *Srsf6* KD RAW MΦ infected with Mtb at (MOI = 5). All data are compared with a SCR control unless indicated. Data are expressed as a mean of three or more biological replicates with error bars depicting SEM. Statistical significance was determined using two tailed unpaired student's *t* test. *=p < 0.05, **=p < 0.01, ***=p < 0.001, ****=p < 0.0001.

suggest that phosphorylation of SRSF6 at S303 promotes splicing of *Bax*, to generate the canonical BAX protein and limit apoptosis.

## Regulation of SRSF6 controls infection outcomes in macrophages

Our data support a model wherein SRSF6 maintains cellular homeostasis and basal type I IFN expression by controlling the abundance of BAX isoforms. Therefore, we predicted that expression of SRSF6 itself could be subject to regulation downstream of pathogen sensing, as a way for cells to prime apoptotic cell death and/or cytosolic DNA sensing via mtDNA release. To further appreciate how macrophages regulate SRSF6 activity, we measured *Srsf6* transcript levels during infection and in response to immune agonists. We consistently detected an approximately 2-fold decrease in *Srsf6* transcript abundance at various time points following *M. tuberculosis* infection of macrophages, as well as in the lung homogenates collected from Mtb-infected mice (day 21 and 77 post-infection) (*Figure 7A–B*). Macrophages sense Mtb infection via several pattern recognition receptors, including dsDNA via cGAS and autocrine sensing of IFN-β via IFNAR (*Watson et al., 2015*). Direct stimulation of these pathways via transfection of dsDNA (*Figure 7C*) or treatment with recombinant IFN-β (rIFN-β) (*Figure 7D*) also led to downregulation of *Srsf6* transcript abundance, as did infection with the gram-negative bacterial pathogen *Salmonella enterica* serovar Typhimurium (*Figure 7E*), engagement of TLR-4 via LPS (*Figure 7F*) and infection with the RNA virus VSV (*Figure 7G*). These data suggest that downregulation of *Srsf6* mRNA levels is part of the general macrophage pathogen sensing response.

Last, we set out to investigate whether SRSF6 plays a more direct role in dictating the outcome of infection with an intracellular pathogen like Mtb. To this end, we infected monolayers of SCR and *Srsf6* KD RAW MΦ cells with a luciferase-expressing strain of Erdman Mtb (MOI = 1) as in *Bell et al., 2021*; *Hoffpauir et al., 2020* and monitored relative light units (RLU) as a measurement of Mtb replication. Remarkably, we found that Mtb grew significantly better in *Srsf6* KD cell lines compared with SCR controls (*Figure 7H*). This increase in Mtb replication was concomitant with hyperinduction of *Ifnb1* and interferon stimulated genes, consistent with high basal type I IFN potentiating type I IFN responses (*Figure 7I*; *West et al., 2015*). Along with harboring higher Mtb bacterial burdens, more *Srsf6* KD RAW MΦ cells stained PI + compared with SCR controls over a 40+h Mtb infection (*Figure 7J*). Collectively, these results demonstrate a critical role for SRSF6 in controlling inflammatory gene expression and cell death during bacterial infection and suggest that regulation of SRSF6, both at the post-translational and transcriptional levels, constitutes an unappreciated layer of complexity in the macrophage innate immune response.

## Discussion

While advances in bioinformatics have facilitated computational detection of alternatively spliced transcripts, assigning function to individual protein isoforms largely remains uncharted territory. Here, we identify the splicing factor SRSF6 as a primary regulator of *Bax* pre-mRNA splicing in murine macrophages. We show that SRSF6-dependent control of the ratio of BAX to BAX-κ isoforms balances critical aspects of innate immune homeostasis in both macrophage-like cell lines and primary macrophages. Specifically, we demonstrate that expression of BAX-κ promotes upregulation of basal type I IFN expression driven by cytosolic mtDNA and renders cells prone to apoptotic cell death. Unlike BAX, which requires a signal to direct mitochondrial targeting and pore formation, BAX-κ is sufficient to trigger type I IFN expression and cell death in resting macrophages (*Figure 5B–D*). These findings position BAX-κ as a potential uncoupler of immunostimulatory mitochondrial DAMP release from cell death and suggests that BAX-κ plays a role in promoting the primed antiviral state that innate immune cells, such as macrophages, must maintain. While we cannot completely rule out the potential for other alternatively spliced genes to contribute to *Srsf6*-dependent innate immune and mitochondrial phenotypes, our data support a significant role for *Bax*-κ in these phenomena.

Connections between SRSF6 and alternative splicing of the cell death protein BAX complement multiple studies that correlate altered SRSF6 expression levels with cancer progression. Our data show that in macrophages, loss of SRSF6 upregulates the pro-apoptotic BAX-κ isoform suggesting that under normal physiological conditions, SRSF6 is a negative regulator of cell death. Consistent with this, upregulation of SRSF6 promotes proliferation of breast cancer cells (*Park et al., 2019*), colorectal cancer cells (*Wan et al., 2019*), and lung cancer cells (*Cohen-Eliav et al., 2013*). Our finding

that loss of SRSF6 expression impacts homeostatic levels of IFN-β and ISGs motivates future studies of SRSF6-dependent immune dysregulation in cancer, particularly in cancers of myeloid cells. Perhaps dysregulation of the immune milieu is part of why SRSF6 upregulation is associated with poor cancer outcomes. At the same time, SRSF6 can also promote expression of pro-apoptotic forms of proteins like BIM (BIM-S) (*Hara et al., 2013*) and cassette exon inclusion in the apoptotic factor FAS (*Choi et al., 2022*). These seemingly contradictory roles for SRSF6 in both limiting and promoting apoptotic cell death suggest that its activity could be concentration and/or cell-type dependent.

Our report that phosphorylation of SRSF6 at S303 promotes *Bax* splicing and limits BAX-κ-dependent apoptotic cell death adds to a growing literature of regulation of gene expression via post-translational modification of splicing factors. Zn(2+)-dependent phosphorylation of SRSF6 has previously been associated with apoptosis via generation of BIM-S (*Hara et al., 2013*) and ubiquitin-mediated control of SRSF6 protein levels has been linked to exon skipping in T cell acute lymphoblastic leukemia (*Zhou et al., 2020*). Our data support a model whereby SRSF6 is regulated at multiple levels downstream of pathogen sensing, through decreasing transcript abundance (*Figure 7*) and differential phosphorylation (*Budzik et al., 2020*). These multiple levels of regulation argue that SRSF6 is a *bona fide* player in the macrophage innate immune response acting as a common signaling molecule to promote a state of antiviral readiness while regulating apoptotic vs. pro-inflammatory cell death. Determining the precise signals that trigger the regulation of SRSF6 in the nucleus downstream of pattern recognition receptors in the plasma membrane and cytosol remain important outstanding questions. Likewise, future experiments that introduce SRSF6 serine mutations directly into the genome will provide additional insight into how post-translational modification alters SRSF6 function in resting vs. activated macrophages.

Seminal work from White et al. and Rongveux et al. implicates apoptotic caspases in suppressing cGAS-dependent type I IFN expression via mtDNA released by BAX/BAK pores during mitochondrial apoptosis (*Rongvaux et al., 2014*; *White et al., 2014*). One interpretation of their findings is that cells prevent aberrant type I IFN expression by dying; thus, one cannot capture *Ifnb1* expression triggered by BAX/BAK pore formation without inhibiting or genetically ablating caspases. However, our work and that of others, begin to suggest that BAX/BAK release of mtDNA and apoptotic cell death can in fact be uncoupled (*Riley et al., 2018*). While our RAW 264.7 macrophage cell lines are selected for stable KD of SRSF6, only a low percentage of resting KD cells stain PI+ (approximately 5–20%) (*Figure 4*). This suggests that some threshold of BAX-κ expression needs to be reached before cells undergo cell death. Because our phenotypes are tightly correlated with the degree of *Srsf6* KD, we predict this threshold is maintained in large part by SRSF6 (*Figure 1F–G*). Curiously, our data also show that cell death in *Srsf6* KD macrophages is caspase-independent and is not concomitant with cytochrome c release (*Figure 4H–I*). While additional experiments at the single cell level are needed to confirm that *Srsf6* KD cells can continuously release mtDNA without releasing cytochrome c and triggering apoptosis, the fact that we were able to uncover the *Srsf6* KD basal type I IFN phenotype in the absence of caspase inhibition argues that some degree of uncoupling is indeed at play.

Consequently, regulation of BAX/BAK pore formation may constitute an unappreciated way for cells to maintain homeostatic levels of IFN signaling, prime antiviral responses, and regulate cell death programs. Indeed, there is growing appreciation that the nature of BAX/BAK pores dictates their ability to release mtDNA and trigger cytosolic DNA sensing. Large BAX/BAK 'macropores' that form at late stages of apoptosis have been shown to allow herniation and extrusion of the inner membrane, releasing mtDNA from the matrix (*McArthur et al., 2018*). The availability of BAX and BAK and the composition of BAX/BAK pores (BAX vs. BAK vs. BAX/BAK) also manages the rate at which pores extrude mtDNA. Specifically, BAK pores are more immunogenic than BAX pores and enable faster mtDNA release (*Cosentino et al., 2022*). It is tempting to speculate BAX-κ promotes mtDNA release and type I IFN expression by disrupting the balance of BAX and BAK at the mitochondria. This could occur by BAX-κ binding to and sequestering BAX away from BAK, allowing for more BAK pore formation. Alternatively, BAX-κ itself could form pores with BAX and/or BAK that preferentially release mtDNA or BAX-κ could form homo-oliogomeric pores with their own unique properties. Detailed visualization of BAX/BAK pores and mtDNA extrusion in cells genetically engineered to express a single BAX isoform (BAX or BAX-κ) alongside biochemical studies of BAX-κ oligomerization will provide important insights into the properties of this isoform and how it contributes to the innate immune phenotypes we report here.

Although we are unable to distinguish between BAX and BAX-κ proteins, our flow cytometry experiments with mitoTRACKER stained mitochondria suggest that BAX-κ is constitutively targeted to mitochondrial membranes (*Figures 4J and 5F*). Alternatively, it is possible that expression of BAX-κ impacts activation or targeting of normal BAX to mitochondria, perhaps by sequestering a negative regulator of BAX (e.g. BCL2). Future experiments using reagents designed to distinguish the two protein isoforms from each other will help resolve these different models of BAX-κ action in macrophages.

Our finding that the intracellular bacterial pathogen *M. tuberculosis* replicates more efficiently in *Srsf6* KD macrophages suggests that maintaining the balance of BAX vs. BAX-κ is needed to restrain bacterial replication and spread. Numerous reports demonstrate that cell death modality usage is a critical factor in determining Mtb pathogenesis. The established paradigm in the Mtb field asserts that apoptosis controls Mtb replication and spread while necrotic cell death promotes it (*Abarca-Rojano et al., 2003*; *Abebe et al., 2011*; *Aguilo et al., 2013*; *Aguiló et al., 2014*; *Behar et al., 2010*; *Behar et al., 2011*). It is curious then, that we observe more Mtb replication and more cell death in *Srsf6* KD RAW MΦ (*Figure 7H and J*). It is possible that SRSF6 contributes to Mtb restriction in BAX-independent ways. It is also possible that basal IFN expression in *Srsf6* KD RAW MΦ creates an intracellular milieu that is ill-adapted for destroying intracellular bacteria (but highly efficient at restricting viral replication *Figure 1P*). Further studies into the precise nature of the cell death that occurs in Mtb-infected *Srsf6* KD RAW MΦ and its reliance on *Ifnb1* may provide important insights into how BAX isoform usage impacts cell-intrinsic control and cell-to-cell spread of pathogens like Mtb.

Collectively, this work highlights the role of pre-mRNA splicing in shaping the macrophage proteome and demonstrates how disruption of protein isoform stoichiometry can impact mitochondrial homeostasis and the ability of cells to respond to infection.

## Materials and methods
### *Mycobacterium tuberculosis*
The Erdman strain was used for all *M. tuberculosis* (Mtb) infections as well as a luciferase expressing strain. Low passage lab stocks were thawed for each experiment to ensure virulence was preserved. Mtb was cultured in roller bottles at 37°C in Middlebrook 7H9 broth (BD Biosciences, GTIN00382902713104) supplemented with 10% OADC (BD Biosciences, 212351), 0.5% glycerol (Fisher, G33-1), and 0.1% Tween-80 (Fisher, BP338-500). All work with Mtb was performed under Biosafety level 3 containment using procedures approved by the Texas A&M University Institutional Biosafety Committee.

### *Salmonella enterica* (ser. Typhimirium)
*Salmonella enterica* serovar Typhimurium (SL1344) was obtained from Dr. Denise Monack at Stanford University. S. T. stocks were streaked out on LB agar plates and incubated at 37 °C overnight.

### Vesicular stomatitis virus
Recombinant Vesicular stomatitis virus (VSV; Indiana serotype) containing a GFP reporter cloned downstream of the VSV G-glycoprotein (VSV-G/GFP) was originally obtained from Dr. John Rose at Yale School of Medicine and shared with us by Dr. A. Phillip West at Texas A&M Health Science Center.

### *Mus musculus*
Mice used in this study were C57BL/6 J (Stock #00064) initially purchased from Jackson labs and afterward maintained with filial breeding. All mice used in experiments were compared with age- and sex- matched controls. Randomization and blinding for in vivo infections was performed by identifying mice based by cage number and ear punch prior to genotyping. Genotype was kept coded until after downstream analysis. Littermates were used for experiments. Mice used to generate BMDMs were males between 10 and 16 weeks old. For in vivo infection male and female mice were infected with Mtb at 10–12 weeks. Embryos used to make primary MEFs were 14.5 days post-coitum. All animals were housed, bred, and studied at Texas A&M Health Science Center under approved Institutional Care and Use Committee guidelines. All experiments for this study were reviewed and approved by the Texas A&M University Institutional Animal Care and Use Committee (AUP# 2019–0083). Mice were fed 4% standard chow and were kept on a 12 hr light/dark cycle and provided food and water ad

libitum. Mice displaying known C57BL/6 J defects were excluded prior to experimentation. Mice were group housed (maximum 5 per cage) by sex on ventilated racks in temperature-controlled rooms.

## Primary cell culture

Bone marrow derived macrophages (BMDMs) were differentiated from bone marrow (BM) cells isolated by washing mouse femurs with 10 mL DMEM 1 mM sodium pyruvate (Thermo Fisher, 11995065). Cells were then centrifuged for 5 minutes at 400 rcf and resuspended in BMDM media DMEM, 20% FBS (Millipore, F0926), 1 mM sodium pyruvate (Lonza, BE13-115E), 10% MCSF conditioned media (Watson lab). BM cells were counted and plated at 5x106 cells per 15 cm non-tissue culture treated dishes in 30 mL complete BMDM media. Cells were fed with an additional 15 mL of BMDM media on day 3. Cells were harvested on day 7 with 1 X PBS EDTA (Lonza, BE02-017F).

Mouse embryonic fibroblasts (MEFs) were isolated from embryos. Briefly, embryos were dissected from yolk sacs, washed 2 times with cold 1 X PBS, decapitated, and peritoneal contents were removed. Headless embryos were disaggregated in cold 0.05% trypsin-EDTA (Lonza, CC-5012) and incubated on ice for 20 minutes, followed by incubation at 37°C for an additional 20 minutes. Cells were then DNase treated with 4 mL disaggregation media (DMEM, 10% FBS, 100 µg/mL DNASE I [Worthington, LS002173]) for 20 minutes at 37°C. Isolated supernatants were spun down at 1000 rpm for 5 minutes. Cells were resuspended in complete MEF media (DMEM, 10% FBS, 1 mM sodium pyruvate), and plated in 15 cm tissue culture-treated dishes 1 dish per embryo in 25 mL of media. MEFs were allowed to expand for 2–3 days before harvest with 0.05% trypsin-EDTA (Lonza, CC-5012).

## Cell culture

RAW 264.7 macrophages (RAW MΦ) (ATCC, TIB-71, originally isolated from male BALB/c mice), were minimally passaged to maintain genomic integrity and new cell lines were generated from low passage stocks. L929 (ATCC, CCL-1) ISRE reporter and cGAS knockout RAW MΦ cell lines have already been produced and described in *Hoffpauir et al., 2020*; *Wagner et al., 2021*, where only low passage cells were used. Cell lines were cultured at 37°C with a humidified atmosphere of 5% $CO_2$ in complete media containing high glucose, DMEM (Thermo Fisher, 11965092) with 10% FBS (Millipore, F0926) 0.2% HEPES (Thermo Fisher, 15630080). All cell lines tested negative for presence of mycoplasma via the Universal Mycoplasma Detection Kit (ATTC, 30–1012 K).

## shRNA knockdowns

For RAW MΦ stably expressing scramble knockdown and Srsf6 knockdown, lentivirus was made by transfection with a pSICO scramble non-targeting shRNA construct and pSICO *Srsf6* shRNA constructs targeted at exon 3 and exon 4 of *Srsf6* using polyjet (SignaGen Laboratories, SL100688). Virus was collected 24 and 48 hr post transfection. RAW MΦ were transduced using lipofectamine 2000 (Thermo Fischer, 52887). After 48 hr, media was supplemented with hygromycin (Invitrogen, 10687010) to select for cells containing the shRNA plasmid.

## siRNA knockdowns

Knockdown of mRNA transcripts was performed by plating $3\times10^5$ RAW MΦ, $3.5\times10^5$ BMDMs on day 4 of differentiation, or $2\times10^5$ MEFs, in 12-well plates and rested overnight. The following day complete media was replaced with 500 µL fresh complete media 30 minutes prior to transfection. Cells were transfected using Fugene SI reagent (SKU:SI-100) or Viromer Blue reagent (VB-01LB-0) and 10 µM of Ambion siRNA stock against either Srsf6 (4390771, IDS86053) or Bax (AM16708, ID100458). For a negative control, Silencer Select Negative Control #1 (Ambion, 4390843) was used. Cells were incubated for 48–72 hr in transfection media at 37°C with 5% $CO_2$ prior to downstream experiments.

## Doxycycline inducible cell line generation

For generation of doxycycline inducible RAW MΦ, pLenti CMV rtTA3 Blast (Addgene, w756-1) stably expressing clonal RAW MΦ were transduced with pLenti CMV Puro DEST (Addgene, w118-1) constructs containing GFP-Strep, Bax201-Strep, Bax203-Strep, BaxG179P-Strep, GFP-FL, SRSF6-FL, and all SRSF6-FL phosphorylation mutants. After 48 hr, construct containing cells were selected through addition of puromycin (Invivogen, ant-pr-1). 1 mg/mL doxycycline (Calbiochem, 324385) treatment was used to activate construct expression.

### *In vitro* infections

#### Mtb infections

To prepare the inoculum, bacteria were grown to mid log phase (OD 0.6–0.8), spun at low speed (500 rcf) to remove clumps, and then pelleted and washed with PBS 2 X. Resuspended bacteria were sonicated and spun at low speed again to further remove clumps. The Mtb was then diluted in DMEM plus 10% horse serum (Gibco, 16050–130) and added to cells at a multiplicity of infection (MOI) of 10 for RNA and protein analysis, a MOI of 5 for cell death studies, and a MOI of 1 for bacterial growth assays. The day before the infection, BMDMs were seeded at $3 \times 10^5$ cells per well (12-well dish), and RAW MΦ were plated on 12-well tissue culture–treated plates at a density of $3 \times 10^5$ cells per well or plated in corning 96 well black plates (CLS3603) at $2.5 \times 10^4$ cells/well and allowed to rest overnight. Cells were spun with bacteria for 10 minutes at 1000 rcf to synchronize infection, washed 2 X with PBS, and then incubated in fresh media. Where applicable, RNA was harvested from infected cells using 0.5 mL TRIzol reagent (Invitrogen, 15596026) at each time point. Protein lysates were harvested with 1 X RIPA lysis buffer and boiled for 10 minutes. For cell death studies propidium iodide (PI) (Invitrogen, P1304MP) was added to media and PI incorporation was measured by fluorescence over time using Lionheart XF analyzer. For bacterial growth assays RAW MΦ were plated on 12-well tissue culture–treated plates at a density of $2.5 \times 10^5$ cells per well. Luminescence was read for *M. tuberculosis* luxB-CADE by lysing in 250 μL 0.5% Triton X-100 and dividing sample into duplicate wells of a 96-well white bottomed plate (Costar, 3693). Luminescence was measured and normalized to background using the luminescence feature of the INFINITE 200 PRO (Tecan) at 0, 48, 72, and 92 hr post infection.

### *Salmonella enterica* (ser. Typhimirium) infections

For *S. enterica* Typhimurium infection, overnight cultures of bacteria were grown in LB broth containing 0.3 M NaCl and grown at 37°C until they reached an OD600 of approximately 0.9 RAW MΦ were seeded in 12-well tissue culture-treated plates at a density of $7 \times 10^5$ cells per well 16 hr before infection. On the day of infection cultures were diluted 1:20. Once cultures had reached mid-log phase (OD600 0.6–0.8) at 2–3 hr, 1 mL of bacteria were pelleted at 5000 rpm for 3 minutes and washed 2 X with PBS. Bacteria were diluted in serum-free DMEM (DMEM Thermo Fisher, 11965092) and added to cells at multiplicity of infection (MOI) of 5. Infected cells were spun at 1000 rpm for 5 minutes then incubated for 10 minutes at 37°C prior to adding fresh media. At indicated times post infection, cells were harvested with TRIzol for RNA isolation described below.

### Vesicular stomatitis viral (VSV) infections

RAW MΦ were seeded in 12-well tissue culture-treated plates at a density of $7 \times 10^5$ cells per well 16 h before infection. The next day cells were infected with VSV-GFP virus at multiplicity of infection (MOI) of 1 in serum-free DMEM (DMEM Thermo Fisher, 11965092). After 1 hr of incubation with media containing virus, supernatant was removed, and fresh DMEM plus 10% FBS (Millipore, F0926) was added to each well. At indicated times post infection, cells were harvested with TRIzol for RNA isolation described below.

### *In vivo* Mtb infections

All infections were performed using procedures approved by Texas A&M University Institutional Care and Use Committee (AUP# 2021–0133). The Mtb inoculum was prepared as described above. Age- and sex-matched mice were infected via inhalation exposure using a Madison chamber (Glas-Col) calibrated to introduce 100–200 bacilli per mouse. For each infection, approximately 5 mice were euthanized immediately, and their lungs were homogenized and plated to verify an accurate inoculum. Infected mice were housed under BSL3 containment and monitored daily by lab members and veterinary staff. At the indicated time points, mice were euthanized, and tissue samples were collected. For cytokine transcript analysis, lungs were homogenized in 500 μL TRIzol, and RNA was isolated as described below.

### Cell stimulation for immune activation and cell death

For immune activation RAW MΦ were plated on 12-well tissue-culture treated plates at a density of $7.5 \times 10^5$ and allowed to rest overnight. Cells were then treated with lipopolysaccharide (LPS) from *E. coli* (Invivogen, tlrl-pb5lps) at 100 ng/mL, or ISD (IDT, annealed in house) at 1 μg/mL, or IFN-β (PBL

 Research article

Chromosomes and Gene Expression | Immunology and Inflammation

Assay Science, 12405–1) at 200 I/U per mL for the respective time points. Cells were collected for RNA isolation using TRIzol reagent. For IFNβ neutralization assays RAW MΦ *Srsf6* KD and SCR cells were treated for 24 hr with IFN-β neutralizing antibody (BD Biolegend, 581302 [1:250]) prior to harvest with TRIzol. For cell death assays RAW MΦ were plated on corning black 96-well half bottom plates at a density of $2.5 \times 10^4$ cells per well and allowed to rest overnight. Complete media was exchanged for complete media containing 5 µg/mL PI and 1 uM staurosporine (Tocaris Bioscience, 1285) or IFN-β neutralizing antibody (BD Biolegend, 581302 [1:250]). Total cell numbers were determined using NucBlue (Thermo Fisher, R37605, [2 drops per mL]) in PBS with a subset of the plated cells. Nuclear incorporation of PI was measured by fluorescence at 4 X magnification using a LionheartXF plate reader (BioTek) every 40 minutes for 24 hr with 5% $CO_2$ at 37°C. For image analysis Gen 3.5 software (BioTek) was used.

## mtDNA depletion
For 2′,3′-dideoxycytidine (ddC) (Abcam, Ab142240) depletion of mitochondrial DNA, RAW MΦ were treated with 10 µM ddC and RNA was harvested after 8 days of culture with TRIzol.

## Seahorse metabolic assays
Seahorse XF mito stress test kits and cartridges were prepared per Agilent's protocols and analyzed on an Agilent Seahorse XF 96-well analyzer. The day before the assay *Srsf6* knockdown RAW MΦ were seeded at $5 \times 10^4$ cells per well and rested overnight. Cells were processed per manufacturer's directions and analyzed using the Agilent Seahorse Mito Stress Test kit (Agilent, 103015–100). Normalization was performed based on absorbance (Abs 562) of total protein concentration measured using a bicinchoninic acid assay (BCA) (Thermo Fisher Scientific, 23225). WAVE software was used for post-acquisition analysis.

## RNA sequencing and analysis
RNA-seq analysis was performed on RAW MΦ containing shRNA knockdowns of *Srsf6*, *Srsf2*, *Srsf1*, *Srsf7*, and *Srsf9* compared with SCR control with biological triplicates of each cell line as described in *Wagner et al., 2021* using Qiagen CLC Genomics software for analysis.

## Alternative splicing analysis
Alternative splicing events were analyzed using Modeling Alternative Junction Inclusion Quantification (MAJIQ) and VOILA (a visualization package) with the default parameters described by *Vaquero-Garcia et al., 2016*. Uniquely mapped, junction-spanning reads were used by MAJIQ to construct splice graphs for transcripts by using the RefSeq annotation supplemented with de-novo detected junctions (de-novo refers to junctions that were not in the RefSeq transcriptome database but had sufficient evidence in the RNA-Seq data). The resulting gene splice graphs were analyzed for all identified local splice variations (LSVs). For every junction in each LSV, MAJIQ quantified expected percent spliced in (PSI) value in control and knockdown samples and expected change in PSI (ΔPSI) between control and knockdown samples. Results from VOILA were then filtered for high confidence changing LSVs (whereby one or more junctions had at least a 95% probability of expected ΔPSI of at least an absolute value of 10 PSI units between control and knockdown) and candidate changing LSVs (95% probability, 10% ΔPSI). For these high confidence results (ΔPSI 10%), the events were further categorized as single exon cassette, multi-exon cassette, alternative 5′ and/or 3′ splice site, or intron-retention (*Wagner et al., 2021*).

## ELISA
siRNA knockdown RAW MΦ were plated in 12-well tissue culture-treated plates at $7.5 \times 10^5$ cells per well and rested overnight. The next day, plates were treated with 10 ng/mL LPS for 3 hr (Invivogen, tlrl-pb5lps) followed by 1 µg/mL poly dA:dT (Invivogen, tlrl-patn-1) or only the 10 ng/mL LPS for 4 hr, and supernatants and RNA was collected using TRIzol. Cytokine levels of IL-1β, were determined using DuoSet ELISA Development Systems (R&D Systems, DY401-05) per manufacturer's protocol using the undiluted cell culture supernatants.

## Flow cytometry

For cell death and apoptosis assays, RAW MΦ were plated in 12-well tissue culture-treated plates at $7.5 \times 10^5$ cells per well and rested overnight. The next day cells were stimulated with 10 µM etoposide (TCI, E0675), or 1 µM ABT737 (Santa Cruz Biotechnology, 207242), or the pan-caspase inhibitor 10 µM Q-VD-OPh (Cayman Chemical, 15260) for the indicated time points prior to being lifted off culture plates with 1 X PBS EDTA (Lonza, BE02-017F). Single cell suspensions were made in 1 X annexin binding buffer. Cells were stained for 5 minutes at RT in 5 µg/mL PI (Invitrogen, P1304MP), and 25 nM annexin-V (APC) (Biolegend, 640912) and were then immediately analyzed on an LSR Fortessa X20 (BD Biosciences). PI fluorescence was measured under PE (585/15). For TMRE assays to assess mitochondrial membrane potential, cells were lifted from culture plates with 1 X PBS EDTA (Lonza, BE02-017F). Single cell suspensions were made in PBS 4% FBS (Millipore, F0926). Cells were stained for 20 minutes at 37°C in 25 nM TMRE (Invitrogen, 11560796), washed 1 X in PBS 4% FBS (Millipore, F0926) and analyzed on an LSR Fortessa X20 (BD Biosciences). Flow-Jo v10software was used for post-acquisition analysis (BD biosciences).

## MitoFLOW

For submicron analysis of mitochondria by flow cytometry, cells were lifted off tissue culture treated plates using 1 X PBS EDTA and added to a 96-well V bottom plate. Cells were pelleted by centrifugation at 400 rcf for 3 minutes. Cells were resuspended in 1 X PBS 2% FBS 200 nM mitoTRACKER green (Invitrogen) and allowed to stain for 15 minutes at 37°C. Cells were then washed once with PBS 2% FBS and were resuspend in ice cold mitoFLOW buffer containing 300 mM sucrose, 10 mM Tris (pH 7.4), 0.5 mM EDTA, and 1 X Halt Protease Inhibitor Cocktail. Lysis was performed by vortexing cells for 3 minutes followed by removal of debris by centrifugation at 400 rcf for 5 minutes at 4°C. For antibody labeling samples were centrifuged at 12,000 rcf for 10 minutes at 4°C and resuspend in 50 µL blocking buffer (5% BSA in mitoFLOW buffer) and incubated on ice for 15 minutes. Blocking was followed by an additional 20 minutes incubation on ice with antibodies of interest (BAX Cell Signaling 2772 S 1:200, Alexa Fluor-647 1:2000 Invitrogen). Mitochondria were washed 2 times in mitoFLOW buffer and were analyzed on an LSR Fortessa X20 (BD Biosciences). Flow-Jo software was used for post-acquisition analysis mitoTRACKER green was used to gate on mitochondria.

## Cytoplasmic DNA enrichment

$7 \times 10^6$ RAW MΦ were plated in 15 cm tissue culture-treated dishes and incubated at 37°C with 5%$CO_2$. The next day cells were lifted with 1 X PBS EDTA (Lonza, BE02-017F) and resuspended in 5 mL PBS. Total DNA was isolated from 2% of resuspended cells, treated with 25 mM NaOH, boiled for 30 minutes, and then neutralized with 50 mM TRIS pH 8.0. The remainder of the cell suspension was pelleted at 3000 rcf for 5 minutes. Cell pellets were resuspended in 500 µL cytosolic lysis buffer (50 mM HEPES pH 7.4, 150 mM NaCl, 50 µg/mL digitonin, 10 mM EDTA) and incubated on ice for 15 minutes. Cells were spun down at 1000 rcf to pellet intact cells and nuclei that were then used for obtaining the membrane fraction. The supernatant was transferred to a fresh tube and spun down at 15000 rcf to remove additional organelle fragments and transferred to a fresh tube again. Cytosolic protein was obtained by transferring 10% of supernatant to a fresh tube with 6 X sample buffer +DTT and boiled for 5 minutes. Cytosolic DNA was isolated from the remaining supernatant by mixing an equal volume of 25:24:1 phenol: chloroform: isoamyl alcohol, with vigorous shaking followed by centrifugation for 10 minutes at ~21,130 rcf (max speed). The aqueous phase was transferred to a fresh tube and DNA was precipitated by mixing with 300 mM sodium acetate, 10 mM $MgCl_2$, 1 µL glycogen, and 3 volumes of 100% ethanol, and then incubated overnight at –80°C. The precipitated DNA was pelleted by centrifugation at max speed for 20 minutes at 4°C. The pellet was washed with 1 mL of cold 70% ethanol and centrifuged for 5 minutes at max speed, and then the pellet was air dried for 15 minutes at RT. The DNA was resuspended with 200 µL DNase free water. For the mitochondrial membrane fraction, the pellet of intact cells, previously collected, was resuspended in 500 µL membrane lysis buffer (50 mM HEPES pH 7.4, 150 mM NaCl, 1% NP-40), vortexed, and then centrifuged for 3 minutes at 7000 rcf. 50 µL of the cleared lysate was transferred to a fresh tube 6 X sample buffer + DTT was added. Samples were boiled 5 minutes. Immunoblotting was used to check for contaminating mitochondrial proteins in the cytosolic fraction compared with the membrane fraction by probing for mitochondrial ATP5A1. RT-qPCR was performed using total DNA diluted 1:100

and cytosolic DNA diluted 1:2. *Tert* (nuclear DNA), *CytB* and *Dloop* (mtDNA) were measured. Total and cytosolic fractions were normalized to *Tert* to control for variation in cell numbers.

## Extracellular IFN-β assay

Macrophage-secreted type I IFN-β levels were determined using a L929 cells stably expressing a luciferase reporter gene under the regulation of type I IFN signaling pathway (L929 ISRE cells). $5\times10^4$ L929 ISRE cells were seeded in a clear 96-well flat-bottomed plate and incubated at 37 °C with 5% $CO_2$ the previous day. Macrophage cell culture media was collected, diluted 1:5 in complete media and transferred to the L929 ISRE cells and incubated for 6 hr. Cells were washed with PBS, lysed in 30 μL cell culture lysis buffer, and transferred to a white 96-well flat-bottomed plate (Costar, 3693). 30 μL of Luciferase Assay System substrate solution (Promega, E1501) was added to the plate and luminescence read immediately using a Cytation5 plate reader (Biotek).

## UV crosslinking immunoprecipitation

$9\times10^6$ RAW MΦ were seeded in 15 cm tissue culture-treated plates and rested overnight. The next day treated with 1 mg/mL doxycycline for 24 hr. Cells were washed with PBS and UV treated at 2000μJoules x100 with a UVstratalinker1800. Cell pellets were lysed in NP-40 lysis buffer with Pierce EDTA free protease inhibitor (Thermo Scientific, A32965) for 15 minutes and sonicated 3 X for 10 minutes 30 seconds on/off (Biorupter). Lysates were treated with 200 ng/mL RNaseA (Invitrogen, AM2271) and 1 U/mL RQ1 Dnase (Promega, M6101) and then incubated on 3XFLAG beads (Sigma Aldrich, F2426) for 3 hr rotating at 4°C. Bound FLAG was eluted 3 X using 20 μL 5XFLAG peptide (Sigma Aldrich, F4799) by vortexing continuously for 15 minutes. Protein samples were collected and separated as described below and RNA was isolated using ethanol precipitation. Briefly, samples were treated with 0.1%SDS and 0.5 mg/mL proteinase K (Invitrogen, 25530049). Samples were spun down at 10,000 rcf for 5 minutes at 4°C, and the pellet was vortexed with equal parts RNase free water and TRIzol, spun down at 10,000 rcf for 20 minutes at 4°C, and then pellets were stored in 100% ethanol +3 M sodium acetate +1 μL glycogen at –80°C overnight. Samples were spun down at 10,000 rcf for 10 minutes at 4°C and then the pellets were washed with 70% ethanol, spun down again, and then pellets were air dried for 15 minutes and RNA was resuspended in RNase free water. qPCR downstream analysis was performed as described below. No RT control cDNA samples were used to check for DNA contamination.

## Immunofluorescence microscopy

Cells were plated on glass coverslips in 24-well plates. At the designated time points, cells were washed with PBS and then fixed in 4% paraformaldehyde for 10 minutes at 37°C. Cells were washed with PBS 3 X and then permeabilized with 0.2% Triton-X100 (Thermo Fisher, A16046.AP). Coverslips were incubated in primary antibody diluted in PBS +5% non-fat milk +0.2% Triton X-100 (PBS-MT) for 2 hr at RT. Primary antibodies used in this study were TOM20 clone 2F8.1(Millipore Sigma, MABT166, 1:100); ANTI-FLAG M2 (Sigma Aldrich, F3165; RRID:AB_259529, 1:500); β-ACTIN (Abcam, 6276; RRID:AB_2223210, 1:500) and DAPI (Invitrogen, D1306, 1:10,000). Coverslips were then washed 3 X in PBS and incubated in secondary antibody (Invitrogen, A-11034, A21235) diluted in PBS-MT for 1 hr in the dark. Coverslips were then washed twice in PBS and then twice in deionized water. Following washes coverslips were mounted onto glass slides using ProLong Diamond antifade mountant (Invitrogen, P36961). Images were acquired on an Olympus Fluoview FV3000 Confocal Laser Scanning Microscope.

## Protein quantification by immunoblot

Cells were washed with PBS and lysed in 1 X RIPA buffer with protease and Pierce EDTA free phosphatase inhibitors (Thermo Scientific, A32957), with the addition of 1 U/mL Benzonase nuclease (Millipore, 101697) to degrade genomic DNA. Proteins were separated by SDS-PAGE on AnykD mini -PROTEAN TGX precast gel (Biorad) and transferred to 0.45 μm nitrocellulose membranes (Cytiva, 10600041). Membranes were blocked for 1 hr at RT in LiCOR Odyssey blocking buffer (927–60001). Blots were incubated overnight at 4°C with the following primary antibodies: β-ACTIN (Abcam, 6276; RRID:AB_2223210, 1:5000), β-TUBULIN (Abcam, 179513, 1:5000) p-IRF3(S396) (Cell Signaling, 49475, 1:1000); IRF3 (Bethyl, A303-384A-M, 1:1000); ANTI-FLAG M2 (Sigma Aldrich, F3165; RRID:AB_259529,

1:5000); NWSHPQFEK (Genscript, A00626-40,1:5000); CGAS (Cell Signaling, 316595, 1:1000); SRP55 (Bethyl, A303-669A-M, 1:1000); VIPERIN (RSAD2) (EMD Millipore, MABF106, 1:1000); ATP5A1 (Bethyl, A304-940A-T, 1:1000); CYTOCHROME-C (Abcam, 133504, 1:1000). Membranes were washed 3 X for 5 minutes in PBS-Tween20 and incubated with appropriate secondary antibodies (LI-COR, 925–32210, 926–68071) for 1 hr at RT prior to imaging on a LiCOR Odyssey Fc Dual-Mode Imaging System.

### RNA isolation and qRT-PCR analysis

For transcript analysis, cells and tissue were harvested in TRIzol and RNA was isolated using Direct-zol RNA Miniprep kits (Zymo Research, R2052) with 1 hr DNase treatment. cDNA was synthesized with iScript cDNA Synthesis Kit (Bio-Rad, 1708891). CDNA was diluted to 1:20 for each sample. A pool of cDNA from each treated or infected sample was used to make a 1:10 standard curve with each standard sample diluted 1:5 to produce a linear curve. RT-qPCR was performed using Power-Up SYBR Green Master Mix (Thermo Fisher, A25742) using a Quant Studio Flex6 (Applied Biosystems). Samples were run in triplicate wells in a 384-well plate. Averages of the raw values were normalized to average values for the same sample with the control gene, *Actb*. To analyze fold induction, the average of the treated sample was divided by the untreated control sample, which was set at 1.

### Semiquantitative PCR analysis

cDNA was synthesized by iScript cDNA Synthesis Kit (Bio-Rad, 1708891) using an extended 3 hr amplification. Q5 high fidelity 2 X Master mix (New England Biolabs, M0492S) was used for PCR amplification using targeted primers. Loading dye was added to PCR products and samples were run on 2% agarose gel containing ethidium bromide (Sigma Aldrich, E1510) at 100 volts for 1 hr. Gels were imaged on LiCOR Odyssey Fc Dual-Mode Imaging System and bands were quantified.

### Quantitation and statistical analysis

Statistical analysis of data was performed using GraphPad Prism software. Two-tailed unpaired Student's t tests were used for statistical analyses based on the assumption that samples were independent and drawn from normal distributions. Unless otherwise noted, all results are representative of at least three biological samples (mean +/- SEM [n=3 per group]). Biological samples refer to independent cell populations. Agarose gel images for semi-quantitative RT-PCR and immunoblots are representative of n>3. For *in vivo* data, we have standardized our sample sizes (number of animals) based on the estimate that detecting a significant effect requires two samples to differ in CFUs by $0.7e^{10}$. Using a standard deviation of $0.35e^{10}$ for each population, we calculated that a minimum size of 5 age- and sex-matched mice per group per time point is necessary to detect a statistically significant difference by a *t*-test with alpha (2-sided) set at 0.05 and a power of 80%. Therefore, we used a minimum of 5 mice per genotype per time point to assess infection-related readouts. For statistical comparison, each experimental group was tested for normal distribution. Data were tested using a Mann-Whitney test. Graphs were generated using Graphpad Prism software.

### Materials availability

The datasets referenced in this study can be found online at National Center for Biotechnology Information (NCBI) Gene Expression Omnibus (GEO), https://www.ncbi.nlm.nih.gov/geo/query/acc.cgi?acc=GSE171418.

Please contact the corresponding author at kpatrick03@tamu.edu for resource sharing and availability.

## Acknowledgements

We would like to thank Dr. A Phillip West and members of the West lab for providing us with VSV stocks and sharing viral infection protocols. We thank Drs. Jefferey Cox, Bennett Penn, and Jonathan Budzik for generously sharing their Mtb phosphoproteomics datasets with us. We thank Dr. Malea Murphy (TAMU COM Integrated Microscopy and Imaging Laboratory) and Robbie Moore (School of Medicine Analytical Cytometry Core [SMAC]) for their technical assistance and advice. Finally, we would like to thank the members of the Patrick and Watson laboratories for their support and reviews of the manuscript. This work was funded through NIH/NIGMS R35GM133720 (to KLP) and NIH/NIAID R01AI155621 (to ROW).

## Additional information

### Funding

| Funder | Grant reference number | Author |
|---|---|---|
| National Institutes of Health | F31GM143893 | Haley M Scott |
| NIH/NIAID | R01AI155621 | Robert O Watson<br>Kristin L Patrick |
| NIH/NIGMS | R35GM133720 | Kristin L Patrick |

The funders had no role in study design, data collection and interpretation, or the decision to submit the work for publication.

### Author contributions

Allison R Wagner, Conceptualization, Formal analysis, Investigation, Visualization, Methodology, Writing - original draft, Writing - review and editing; Chi G Weindel, Conceptualization, Supervision, Investigation, Visualization, Methodology, Writing - review and editing; Kelsi O West, Haley M Scott, Conceptualization, Investigation; Robert O Watson, Conceptualization, Supervision, Funding acquisition, Visualization, Methodology, Writing - review and editing; Kristin L Patrick, Conceptualization, Supervision, Funding acquisition, Visualization, Writing - original draft, Writing - review and editing

### Author ORCIDs

Allison R Wagner ⓘ http://orcid.org/0000-0002-6592-3741
Chi G Weindel ⓘ http://orcid.org/0000-0002-8063-7794
Robert O Watson ⓘ http://orcid.org/0000-0003-4976-0759
Kristin L Patrick ⓘ http://orcid.org/0000-0003-2442-4679

### Ethics

All experiments for this study were reviewed and approved by the Texas A&M University Institutional Animal Care and Use Committee (AUP# 2019-0083).

### Decision letter and Author response

Decision letter https://doi.org/10.7554/eLife.82244.sa1
Author response https://doi.org/10.7554/eLife.82244.sa2

---

## Additional files

### Supplementary files

• MDAR checklist

### Data availability

Sequencing data have been deposited in GEO under accession code GSE171418. All other data generated or analyzed during this study are included in the manuscript and supporting files.

The following previously published dataset was used:

| Author(s) | Year | Dataset title | Dataset URL | Database and Identifier |
|---|---|---|---|---|
| Wagner AR, Scott HM, West KO, Vail KJ, Fitzsimons TC, Coleman AK, Carter KE, Watson RO, Patrick KL | 2021 | Global transcriptomics reveals specialized roles for splicing regulatory proteins in the macrophage innate immune response | https://www.ncbi.nlm.nih.gov/geo/query/acc.cgi?acc=GSE171418 | NCBI Gene Expression Omnibus, GSE171418 |

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

## Appendix 1

### Appendix 1—key resources table

| Reagent type (species) or resource | Designation | Source or reference | Identifiers | Additional information |
|---|---|---|---|---|
| Strain, strain background (*Vesicular stomatitis virus*, Indiana serotype) | VSV | Dr. John Rose, Yale School of Medicine | | contains a GFP reporter cloned downstream of the VSV G-glycoprotein (VSV-G/GFP) |
| Strain, strain background (*S. enterica* (ser. Typhimirium)) | *Salmonella typhimurium, Sal* | Dr. Denise Monack, Stanford | Cat#SL1344 | |
| Strain, strain background (*Mycobacterium tuberculosis* (Erdman)) | *M. tuberculosis, Mtb* | ATCC | Cat#35801 | |
| Cell line (*M. musculus*) | RAW 264.7 macrophages | ATCC | Cat#TIB-71 | |
| Cell line (*M. musculus*) | Tetracycline Inducible RAW 264.7 | This paper, Dr. Robert Watson, Texas A&M School of Medicine | | contains a reverse tetracycline controlled transactivator and an upstream tetracycline inducible promotor containing plasmid |
| Cell line (*Homo sapien*) | L929 ISRE reporter cells | **Wagner et al., 2021** **Hoffpauir et al., 2020** | | Dr. Robert Watson, Texas A&M School of Medicine |
| Cell line (*M. musculus*) | cGAS KO RAW 264.7 | **Wagner et al., 2021** | | Dr. Robert Watson, Texas A&M School of Medicine |
| Transfected construct (*M. musculus*) | *Srsf6* shRNA | This paper, Dr. Kristin Patrick, Texas A&M School of Medicine | KD1 (exon 3) KD2 (exon 4) | lentiviral plasmid with hygromycin resistance |
| Transfected construct (*M. musculus*) | Negative control (NC) siRNA | Ambion silencer select siRNA | Cat#4390843 | |
| Transfected construct (*M. musculus*) | *Bax* siRNA | Ambion silencer pre-designed siRNA | Cat#AM16708 ID100458 | |
| Transfected construct (*M. musculus*) | *Srsf6* siRNA | Ambion silencer select pre-designed siRNA | Cat#4390771 IDS86053 | |
| Antibody | BAX Rabbit polyclonal | Cell Signaling | Cat#2772 S | (1:1000) (1:200) |
| Antibody | VIPERIN mouse monoclonal | EMD Millipore | Cat#MABF106 | (1:1000) |
| Antibody | SRp55 Rabbit polyclonal | Bethyl | Cat#A303-669A-M | (1:1000) |
| Antibody | p-IRF3(S396) Rabbit monoclonal | Cell Signaling | Cat#49475 | (1:1000) |
| Antibody | IRF3 Rabbit polyclonal | Bethyl | Cat#A303-384A-M | (1:1000) |
| Antibody | cGAS Rabbit monoclonal | Cell Signaling | Cat#316595 | (1:1000) |
| Antibody | Tom20/Tomm20 mouse monoclonal clone 2F8.1 | EMD Millipore | Cat#MABT166 | (1:1000) |
| Antibody | VDAC1 Mouse monoclonal clone N152B/23 | Biolegend | Cat#820702 | (1:1000) |
| Antibody | Cytochrome C monoclonal Rabbit | Abcam | Cat#133504 | (1:1000) |
| Antibody | (Strep)NWSHPQFEK Rabbit polyclonal | GenScript | Cat#A00626-40 | (1:5000) |
| Antibody | FLAG M2 Mouse Monoclonal | Sigma-Aldrich | Cat#F3165; RRID:AB_259529 | (1:5000) |
| Peptide, recombinant protein | Recombinant mouse IFN-β1 (carrier free) | Biolegend | Cat#581302 | |
| Peptide, recombinant protein | Recombinant IFNβ | PBL Assay Science | Cat#12405–1 | |
| Peptide, recombinant protein | *E. coli* Lipopolysaccharide (LPS) | InvivoGen | Cat# tlrl-pb5lps | |
| Peptide, recombinant protein | Interferon stimulatory DNA (ISD) | IDT | | |
| Sequence-based reagent | *Srsf6*_F | This paper, Dr. Kristin Patrick, Texas A&M School of Medicine | qRT-PCR primer | GACATCCAGCGC TTTTTCAG |

*Appendix 1 Continued on next page*

*Appendix 1 Continued*

| Reagent type (species) or resource | Designation | Source or reference | Identifiers | Additional information |
|---|---|---|---|---|
| Sequence-based reagent | Srsf6_R | This paper, Dr. Kristin Patrick, Texas A&M School of Medicine | qRT-PCR primer | TTGAGGTCGAT CTCGAGGAG |
| Sequence-based reagent | Rsad2_F | This paper, Dr. Kristin Patrick, Texas A&M School of Medicine | qRT-PCR primer | ATAGTGAGCAAT GGCAGCCT |
| Sequence-based reagent | Rsad2_R | This paper, Dr. Kristin Patrick, Texas A&M School of Medicine | qRT-PCR primer | AACCTGCTCAT CGAAGCTGT |
| Sequence-based reagent | Bax-κ_F | This paper, Dr. Kristin Patrick, Texas A&M School of Medicine | qRT-PCR primer | AGAGGCAGCGGCAGTGAT |
| Sequence-based reagent | Bax-κ_R | This paper, Dr. Kristin Patrick, Texas A&M School of Medicine | qRT-PCR primer | GGGGTCCTAGGGTTCTTGG |
| Sequence-based reagent | Bax_F | This paper, Dr. Kristin Patrick, Texas A&M School of Medicine | qRT-PCR primer | CCGGCGAATTGG AGATGAACTG |
| Sequence-based reagent | Bax_R | This paper, Dr. Kristin Patrick, Texas A&M School of Medicine | qRT-PCR primer | AGCTGCCACCCGG AAGAAGACCT |
| Sequence-based reagent | Bax_F | This paper, Dr. Kristin Patrick, Texas A&M School of Medicine | PCR primer | AGAGGCAGCGGCAGTGAT |
| Sequence-based reagent | Bax_R | This paper, Dr. Kristin Patrick, Texas A&M School of Medicine | PCR primer | CTCAGCCCATCTTCTTCCAG |
| Commercial assay or kit | Luciferase Assay System | Promega | Cat#E1501 | |
| Commercial assay or kit | Direct-zol RNA miniprep Kit | Zymo Research | Cat#R2052 | |
| Commercial assay or kit | Viromer Blue | Lipocalyx | Cat#VB-01LB-0 | |
| Commercial assay or kit | Seahorse XF Cell Mito Stress Test Kit | Agilent | Cat#103015–100 | |
| Chemical compound, drug | Alexa Fluor 647 AnnexinV | Biolegend | Cat#640912 | |
| Chemical compound, drug | Tetramethylrhodamine, ethyl ester (TMRE) | Invitrogen | Cat#11560796 | |
| Chemical compound, drug | Propidium Iodide (PI) | Invitrogen | Cat#P1304MP | |
| Chemical compound, drug | Mitotracker Green FM | Invitrogen | Cat#M7514 | |
| Chemical compound, drug | 2′,3′-dideoxycytidine (DDC) | Abcam | Cat# Ab142240 | |
| Chemical compound, drug | Q-VD-OPh | Cayman chemical | Cat#15260 | |
| Chemical compound, drug | Staurosporine | Tocaris Bioscience | Cat#1285 | |
| Chemical compound, drug | TRIzol | Invitrogen | Cat#15596026 | |
| Software, algorithm | CLC Genomics Workbench 8.0.1 | QIAGEN bioinformatics | https://www.qiagen bioinformatics. com/products/clc-genomics-workbench/ | |
| Software, algorithm | MAJIC & VIOLA | *Vaquero-Garcia et al., 2016* | https://majiq.biociphers. org/ | |
| Software, algorithm | Integrated genomics viewer | Broad Institute | | |
| Software, algorithm | FlowJo v10 | BD biosciences | | |
| Software, algorithm | Prism v7 | Graph Pad | | |

