## [Editor Report]

SR proteins are a family of RNA binding proteins that have widespread essential functions throughout biology. SRSF6 is an understudied SR family member, best characterized for its role in controlling alternative splicing. Through comparative RNA-Seq analysis, this study demonstrates a role of SR protein SRSF6 in regulating interferon-responsive gene expression in macrophages. Moreover, the data provide compelling evidence that SRSF6 influences the interferon response through controlling mitochondrial damage triggered by a spliced isoform of BAX.

---

## [Decision Letter]

**Decision letter after peer review:**

Thank you for submitting your article "SRSF6 balances mitochondrial-driven innate immune outcomes through alternative splicing of BAX" for consideration by *eLife*. Your article has been reviewed by 3 peer reviewers, one of whom is a member of our Board of Reviewing Editors, and the evaluation has been overseen by Satyajit Rath as the Senior Editor. The reviewers have opted to remain anonymous.

Essential revisions:

The reviewers all agree this is generally a strong study but they would like to see some further investigation of at least the first of the following two points:

1) How is Bax involved in mediating the mitochondrial damage? As the reviewers point out, the expression of the kappa isoform seems minimal. This raises questions as to whether the kappa isoform really is doing anything, or is the impact through a reduction in the full length isoform or are other genes also linking SFSR6 depletion to mitochondrial damage.

2) Can the authors provide more specifics on how mitochondrial damage is linked in this case to the cellular phenotypes of innate immune triggering and cell death. In particular, are caspases involved?

*Reviewer #1 (Recommendations for the authors):*

Specific suggestions:

1) To strengthen the conclusion regarding the loss of mitochondrial integrity in response to SRSF6 depletion that authors should minimally quantify the number of cells that exhibit the difference shown in Figure 2G, and directly test if the relatively small change in mitochondrial polarization is sufficient to activate cGAS.

2) To argue that BAX intron retention is relevant to the mitochondrial and ISG phenotypes, the authors need to (a) show that Bax-k is detectable at the protein level in SRSF6 depleted cells and (b) show that expressing only a small amount of Bax-k in the background of full-length Bax (i.e. by ASO or titrating the cDNA expression experiment), as observed upon SRSF6 KD, is sufficient to induce the ISGs.

3) Given the likelihood that BAX splicing does not explain all of the phenotype, it would be of interest to know more about the other mitochondrial genes that have SRSF6-regulated splicing events (i.e. from Figure 3B). How big are these changes? What is the predicted impact on protein expression? Might any of these work synergistically with BAX?

4) Similarly, it is important to know if intron retention of BAX causes a change in the protein expression of any other mitochondrial integrity or cell death proteins.

*Reviewer #2 (Recommendations for the authors):*

1. The data in Figure 3E-F showing "preferential retention of intron 1" is not compelling. In 3F, there doesn't seem to be more Bax-kappa but rather less Bax. This seems important for interpreting the role of Bax-kappa in promoting cell death. This becomes especially important because in WT cells, Bax-k overexpression induced apoptosis is caspase-dependent (unlike in the SF6 KD), as the authors note. In Figure 5E – they argue that is due to overexpression – it would be helpful to gauge expression of endogenous Bax to overexpressed Bax-K (either by immunoblot or qRTPCR). I'm not convinced it is this isoform of Bax that is critical, although the data showing that the SRSF6 KD phenotype depends on SOME isoform of Bax is more convincing.

2. There seem to be some discrepancies in the data from figure to figure. For example:

– Rsad2 expression is significantly increased in the Srsf6 KD in Figure 1, but it is not marked as significantly different in Figure 3L.

– Numbers of PI+ cells are significantly different between SCR and KD in Figure 4A or 4H but that difference doesn't seem to be reflected in the PI+ population in Figure 4G.

3. I really liked the approach of using phospho-dead or phospho-mimetic alleles in Figure 6. Although I can see that the data are statistically significant, the overall difference between the phospho-dead and phospho-mimetic alleles in the cell death assays seems modest.

*Reviewer #3 (Recommendations for the authors):*

In Fig4A, B, SRSF6 KD cells are sensitized to cell death and the basal π signal is elevated relative to control cells. I was expecting the PI+ signal in Figure 4A, B to increase over time as more SRSF6 KD cells, which are sensitized to death, die over time. However, the signal decreases and plateaus. Can the authors explain this? Are you selecting for a population of cells that eventually have high enough levels of SRSF6 such that they are no longer sensitized to cell death?

In Figure 4C, D, the authors observe an increase in both PI+ and Annexin V+ signal, which suggest lytic cell death. They concluded that this cell death was caspase independent as it was not abrogated by the use of Q-VD-OPh, a pan caspase inhibitor. In the discussion they comment that apoptosis restricts *M. tuberculosis* replication, but that necrotic death potentiates replication. Given that the authors observed *M. tuberculosis* replication advantage in SRSF6 KD cells, it suggests that there may be necrotic cell death. However, they never characterize the mechanism of cell death mediated by BAX-kκ biochemically. It would be nice to see western blots for markers of apoptosis such as cleaved PARP and CASP3, as well as markers of lytic death such as GSDMD and GSDME to determine the precise mechanism of death. Additionally, I would suggest using ZVAD-FmK, a pan caspase inhibitor commonly used in the cell death field to test the dependence on caspases. This is an interesting finding and given that not much is known about the caspase independent mitochondria mediate cell death, it's worth the additional characterization, especially since overexpression of BAX-k induced cell death that was inhibited by Q-VD-OPh (Figure 5).

The western blot in Figure 6B depicting the expression of the SRSF6 variants suggests that the phospho mimetic SRSF6 S303D is not expressed yet they claim this phosphorylation event determines the ratios of BAX and BAX-κ to fine tune the sensitivity to apoptosis. This western blot is not convincing. It might also be worthwhile to include an inducible protein control such as GFP (Figure 6D) which would not be expected to alter the ratio of Bax-kto Bax.

---

## [Author Response]

Essential revisions:The reviewers all agree this is generally a strong study but they would like to see some further investigation of at least the first of the following two points:1) How is Bax involved in mediating the mitochondrial damage? As the reviewers point out, the expression of the kappa isoform seems minimal. This raises questions as to whether the kappa isoform really is doing anything, or is the impact through a reduction in the full length isoform or are other genes also linking SFSR6 depletion to mitochondrial damage.

We’ve worked hard over the past couple of months to strengthen our evidence that Bax-k is the primary driver of mitochondrial instability in *Srsf6* KD macrophages. Because Bax-k is a constitutively active form of BAX (supported by our work and through studies of the analogous isoform in humans (Cartron et al., 2005)), it is not surprising that cells don’t express tons of it— any cell that expresses too much Bax-k will likely die. Regardless, we have addressed these valid concerns through changes to the text and through additional experiments, detailed here.

A. First off, acknowledging reviewers concerns that the mitochondrial phenotypes reported could be contributed to by additional SRSF6-dependent splicing changes, we have added text to the manuscript addressing this caveat on line 436.

B. To better link mitochondrial phenotypes to Bax-k, we sorted cells using flow cytometry based on their degree of mitochondrial membrane depolarization (as measured by TMRE staining: less TMRE = more depolarization). We then isolated RNA from these cells and measured Bax-k transcripts by RT-qPCR. We found that cells with more mitochondrial membrane depolarization expressed more Bax-k (new Figure 4K). This was even the case with SCR cells—arguing this isoform is associated with mitochondrial instability in wild-type cells. This finding strengthens the link between expression of the Bax-k isoform and the phenotypes we report in our population of *Srsf6* KD cells.

C. Also using flow cytometry, we applied a new technique (recently published in Weindel et al., 2022), wherein we isolated mitochondria (based on MitoTracker green staining) from macrophages and then measured the amount of a particular protein enriched on these isolated organelles (we call this mitoFLOW). We did this using endogenous antibodies against total BAX, with the prediction that we would see increased BAX associated with mitochondria in cells expressing more Bax-k, as Bax-k is proposed to constitutively associate with mitochondrial membranes. We measured more BAX+ mitochondria in Srsf6 KD macrophages (vs. SCR) (new Figure 4J) and in Bax-k dox-inducible cells (vs. normal BAX dox-inducible cells) (new Figure 5F).

This data suggests that Bax-k promotes mitochondrial instability via its propensity to associate with the mitochondrial outer membrane. Because we cannot distinguish between normal BAX and Bax-k in this experiment, we are aware of the possibility that it is not actually Bax-k on these mitochondria and that somehow Bax-k promotes association of normal BAX (perhaps by sequestering a negative regulator of BAX localization). Either way, our conclusions are valid, but we are careful to propose this alternative scenario in our Discussion (line 496).

D. We also provide additional evidence arguing against *Srsf6* KD phenotypes being the result of loss of normal BAX, including a western blot of total BAX in SCR vs. *Srsf6* KD cell lines (new Figure S3C) and data showing a lack of ISG induction if we knockdown total *Bax* in WT RAW 264.7 cells (new Figure 3L).

2) Can the authors provide more specifics on how mitochondrial damage is linked in this case to the cellular phenotypes of innate immune triggering and cell death. In particular, are caspases involved?

We hypothesized that the difference in caspase-dependence between the *Srsf6* KD cells and cells overexpressing BAX-k was largely due to dose of BAX-k and the length of time that cells were exposed to this harmful variant protein. Caspase-independent cell death, while still poorly defined, is thought to result from prolonged, low-level damage to the mitochondrial network. This is precisely the type of mitochondria stress we’d predict occurs in *Srsf6* KD cells, which are a *stable selected* cell line that we investigate under steady-state conditions. In contrast, cells in the dox system go from expressing normal, low levels of Bax-k to expressing appreciable amounts of it over the course of hours. This scenario represents a more acute response that would be predicted to fully engage the apoptotic cascade.

To attempt to address this experimentally, we decided that expressing as little BAX-k as possible would be the best way to mimic the *Srsf6* KD cell lines. Therefore, we repeated our inducible expression experiment at an early (5h) and late (24h) time point after dox addition, using flow cytometry to measure AnnexinV+ and PI- cells at each time point. Consistent with our prediction, we see a dose dependent increase of dead and apoptotic cells as Bax-k accumulates This cell death is more dependent on caspases at the later time point than at the early time point (new Figure S5D). We also see a dose-dependent increase in ISGs as we turn on Bax-k expression (new Figure 5B). These data support a model whereby low, sustained levels of Bax-k levels stress the mitochondria in a way that leads to caspase-independent cell death but a sudden large increase in Bax-k expression is enough to push cells fully into more canonical apoptosis. Additional experiments outside the scope of this manuscript will pursue whether cell death pathways like necroptosis are engaged downstream of Bax-k-dependent mitochondrial dysfunction (although we do report the inflammasome in likely not involved, see Figure S4C).

Reviewer #1 (Recommendations for the authors):Specific suggestions:1) To strengthen the conclusion regarding the loss of mitochondrial integrity in response to SRSF6 depletion that authors should minimally quantify the number of cells that exhibit the difference shown in Figure 2G, and directly test if the relatively small change in mitochondrial polarization is sufficient to activate cGAS.

Because IFN-b acts in a powerful autocrine fashion through IFNAR, we are unable to specifically demonstrate “cell-intrinsic” cGAS activation in cells with mitochondrial depolarization—all of the *Srsf6* KD cells in a culture will be exposed to higher levels of IFN-b in the supernatants so all cells will express ISGs and have some degree of IRF3/STAT1 activation, etc. But we interpret the heart of your question to be an interest in correlating Bax-k expression with the phenotypes we report in *Srsf6* KD cells. To this end, we used flow cytometry to sort SCR and *Srsf6* KD macrophages based on TMRE signal, a measure of mitochondrial membrane polarization. Then we isolated RNA from cells in each “bin” and measured Baxk transcript by RT-qPCR. As seen in new Figure 4K, we show that Bax-k expression is higher in cells with the greatest loss of membrane potential (lowest TMRE). This finding strengthens the link between expression of the Bax-k isoform and the mitochondrial depolarization phenotypes we report in our population of *Srsf6* KD cells.

2) To argue that BAX intron retention is relevant to the mitochondrial and ISG phenotypes, the authors need to (a) show that Bax-k is detectable at the protein level in SRSF6 depleted cells and (b) show that expressing only a small amount of Bax-k in the background of full-length Bax (i.e. by ASO or titrating the cDNA expression experiment), as observed upon SRSF6 KD, is sufficient to induce the ISGs.

We have definitely tried to detect endogenous Bax-k by western blot and cannot convince ourselves that we are truly visualizing the isoform. Because Bax-k is a constitutively active, proapoptotic protein, we think it makes sense that cells will express a very limited amount of it (any cell expressing too much will die). We know this is the case because our dox-inducible cells expressing Bax-k start to die a few hours post induction. Equally frustrating is the fact that we can’t selectively knockdown Bax-k at the RNA level without potentially impacting full length Bax: the coding sequence of Bax-k is identical to the *Bax* pre-mRNA (they differ only through retention of intron 1 so any RNA-based targeting could impact both isoforms).

Determined to address your concern experimentally, we turned to our inducible Bax/Bax-k system and repeated our “sufficiency” experiments over a time-course of dox induction. Even at early time points post-induction (6h) when Bax-k is just barely detectable, we can start to see accumulation of ISGs and cell death (Figures 5B, S5D). We believe this better recapitulates the situation in *Srsf6* KD cells. Also, showing dose-dependence strengthens our argument that Baxk is the major driver of our phenotypes.

We also repeated our total Bax KD experiment, this time including data to show that knockdown of total Bax only rescues ISG expression in an *Srsf6* KD background (and not in normal WT RAW 264.7 cells) (new Figure 3K-L). This result also helps argue that *Bax* alternative splicing is the major driver of mitochondrial instability and ISG expression in *Srsf6* KD cells.

3) Given the likelihood that BAX splicing does not explain all of the phenotype, it would be of interest to know more about the other mitochondrial genes that have SRSF6-regulated splicing events (i.e. from Figure 3B). How big are these changes? What is the predicted impact on protein expression? Might any of these work synergistically with BAX?

Absolutely, there are many other alternative splicing changes induced by loss of SRSF6 that may contribute to the phenotypes in *Srsf6* KD macrophages. We are of the opinion that the sufficiency experiments in Figure 5 and Bax KD rescue experiments in Figure 3K provide pretty strong support that Bax-k is the major player but we cannot rule out other events contributing. Because experimentally investigating additional splicing events is outside the scope of this manuscript, we have added text to explicitly state that we cannot for sure conclude that Bax-k is the only driver of mitochondrial instability and cell death in *Srsf6* KD macrophages (line 436).

4) Similarly, it is important to know if intron retention of BAX causes a change in the protein expression of any other mitochondrial integrity or cell death proteins.

Good point. We’ve measured the abundance of two additional mitochondrial factors at the protein level (VDAC1 and TOM20) and do not measure major differences in their abundance in SCR vs. *Srsf6* KD. We have added this to Figure S2B.

Reviewer #2 (Recommendations for the authors):1. The data in Figure 3E-F showing "preferential retention of intron 1" is not compelling. In 3F, there doesn't seem to be more Bax-kappa but rather less Bax. This seems important for interpreting the role of Bax-kappa in promoting cell death. This becomes especially important because in WT cells, Bax-k overexpression induced apoptosis is caspase-dependent (unlike in the SF6 KD), as the authors note. In Figure 5E – they argue that is due to overexpression – it would be helpful to gauge expression of endogenous Bax to overexpressed Bax-K (either by immunoblot or qRTPCR). I'm not convinced it is this isoform of Bax that is critical, although the data showing that the SRSF6 KD phenotype depends on SOME isoform of Bax is more convincing.

We agree that the increase in Bax-k transcript expression is modest. Because Bax-k is a constitutively active, pro-apoptotic protein, we think it makes sense that cells will express a very limited amount of it (any cell expressing too much will die). Because each *Bax* pre-mRNA synthesized by RNAPII will either be processed into *Bax* or Bax-k it does conceptually make sense that more Bax-k = less normal *Bax* (ignoring the obvious caveat of mRNA stability), but we do not think that loss of *Bax* drives our phenotypes. We actually tested if *Bax* KD in WT cells could recapitulate phenotypes in *Srsf6* KD cells and observed no change to ISG expression (new data, Figure 3L). Likewise, overexpression of normal BAX in our dox-inducible system had almost no impact on ISG expression or cell death (which makes sense: in the absence of an apoptosis inducing signal, full length BAX is not active) (Figure 5C-D). In new data, using an antibody against all endogenous BAX, we isolated mitochondria and quantified BAX+ mitochondria in SCR and *Srsf6* KD cells (as well as in the BAX/Bax-k inducible cells). We observe a considerable increase in BAX association with mitochondria in *Srsf6* KD cells and in Bax-k expressing cells, compared to SCR or normal BAX expressing cells. This new data supports a model wherein Bax-k is constitutively associated with mitochondrial membranes, resulting in mitochondrial membrane instability and mtDNA leakage.

Re: your concern about the caspase-dependency—this always puzzled us too. We hypothesized that the difference between the *Srsf6* KD phenotype and the overexpression of Bax-k was largely due to dose and timing. Caspase-independent cell death, while still poorly defined, is thought to result from prolonged, low-level damage to the mitochondrial network (Tait and Green 2008). This is precisely the type of mitochondrial stress we’d predict occurs in *Srsf6* KD cells, which are a *stable selected* cell line. In contrast, cells in the dox system go from expressing normal, low levels of Bax-k to expressing appreciable amounts of it over the course of hours. We predicted that expressing as little Bax-k as possible would be the best way to mimic the *Srsf6* KD cell lines, so we repeated our inducible expression experiment at an early (6h) and late (24h) time point after dox addition, using flow cytometry to measure AnnexinV+ and PI- cells at each time point. Consistent with our prediction, we see dose dependent increase of apoptotic cell death as Bax-k accumulates and this cell death is more dependent on caspases at the later time point than at the early time point.

2. There seem to be some discrepancies in the data from figure to figure. For example:– Rsad2 expression is significantly increased in the Srsf6 KD in Figure 1, but it is not marked as significantly different in Figure 3L.– Numbers of PI+ cells are significantly different between SCR and KD in Figure 4A or 4H but that difference doesn't seem to be reflected in the PI+ population in Figure 4G.

We have repeated the experiment in Figure 3L and upon repeat, our cells demonstrated the predicted increase in *Rsad2* increase, which was rescued by knocking down Bax.

We do see some slight variations between overall % PI+ cells (or in some experiments, bulk π signal) between experiments—this is in large part due to whether the experiment was done right after the *Srsf6* KD and SCR cell lines were derived (via lentiviral transduction of stable shRNA constructs) or if it was a week or two after they were derived (more and more PI+ cells accumulate as we passage the *Srsf6* KD cell lines, which is why we only passage them for about 3 weeks before tossing them and starting experiments with freshly derived cells). You can rest assured that all our phenotypes were consistently replicated over and over during the 2+ years we spent on this study.

3. I really liked the approach of using phospho-dead or phospho-mimetic alleles in Figure 6. Although I can see that the data are statistically significant, the overall difference between the phospho-dead and phospho-mimetic alleles in the cell death assays seems modest.

We agree that the phenotypes in the phosphomutant cell lines are modest. But we also like the idea of the macrophage using phosphorylation to regulate SRSF6 function so we chose to include the data, mainly as a starting point for future experiments. Admittedly, our experimental system here is not ideal as we are comparing overexpression of WT SRSF6 to overexpression of the phosphomutants on top of endogenous SR6. Future experiments that probe the function of these phosphorylations in a cell line in which serine mutations have been directly engineered into the genome would most certainly yield clearer results. We explicitly point out the shortcomings of our experimental design in line 460 and how we’d like to pursue it in future.

Reviewer #3 (Recommendations for the authors):In Figure 4A, B, SRSF6 KD cells are sensitized to cell death and the basal π signal is elevated relative to control cells. I was expecting the PI+ signal in Figure 4A, B to increase over time as more SRSF6 KD cells, which are sensitized to death, die over time. However, the signal decreases and plateaus. Can the authors explain this? Are you selecting for a population of cells that eventually have high enough levels of SRSF6 such that they are no longer sensitized to cell death?

Good point! Showing the data in Figure 4A as a time course is confusing. Since no

stimulus/apoptosis inducing agent is added to the cells, we wouldn’t expect an increase in cell death over a 20h time course. These are just our resting *Srsf6* KD cell lines and we’re measuring π incorporation at steady state (after we select them, the population stays pretty happy for about 3 weeks and then we toss them and rederive). We’ve replaced 4A and 4B with data for a single time point, which is less confusing, since we are measuring steady state π levels in both cases.

In Figure 4C, D, the authors observe an increase in both PI+ and Annexin V+ signal, which suggest lytic cell death. They concluded that this cell death was caspase independent as it was not abrogated by the use of Q-VD-OPh, a pan caspase inhibitor. In the discussion they comment that apoptosis restricts M. tuberculosis replication, but that necrotic death potentiates replication. Given that the authors observed M. tuberculosis replication advantage in SRSF6 KD cells, it suggests that there may be necrotic cell death. However, they never characterize the mechanism of cell death mediated by BAX-kκ biochemically. It would be nice to see western blots for markers of apoptosis such as cleaved PARP and CASP3, as well as markers of lytic death such as GSDMD and GSDME to determine the precise mechanism of death. Additionally, I would suggest using ZVAD-FmK, a pan caspase inhibitor commonly used in the cell death field to test the dependence on caspases. This is an interesting finding and given that not much is known about the caspase independent mitochondria mediate cell death, it's worth the additional characterization, especially since overexpression of BAX-k induced cell death that was inhibited by Q-VD-OPh (Figure 5).

Because only a small proportion of *Srsf6* KD cells show signs of cell death at any one time (Figure 4C), we had no luck using bulk assays like western blot to measure cell death pathway events. In the future, we can think about sorting out populations of AnnexinV+/PI- cells to enrich for cells that are actually undergoing this transition.

Re: your concern about the caspase-dependency—this always puzzled us too. We hypothesized that the difference between the *Srsf6* KD phenotype and the overexpression of Bax-k was largely due to dose and timing. Caspase-independent cell death, while still poorly defined, is thought to result from prolonged, low-level damage to the mitochondrial network. This is precisely the type of mitochondria stress we’d predict occurs in *Srsf6* KD cells, which are a *stable selected* cell line. In contrast, cells in the dox system go from expressing normal, low levels of Bax-k to expressing appreciable amounts of it over the course of hours—this probably constitutes are more acute stress to the cells and pushes them fully into caspase-dependent cell death. We predicted that expressing as little Bax-k as possible would best mimic the *Srsf6* KD cell lines so we repeated our inducible expression experiment at an early (6h) and late (24h) time point after dox addition, using flow cytometry to measure AnnexinV+ and PI- cells at each time point. Consistent with our prediction, we see dose dependent increase of apoptotic cell death as Bax-k accumulates and this cell death is more dependent on caspases at the later time point than at the early time point (Figure S5D).

Following your suggestion, we treated *Srsf6* KD and SCR cells with another caspase inhibitor, zVAD-FMK, and found that it also did not rescue cell death in *Srsf6* KD cells, similarly to QVDOPH. We added this data to Figure S4D.

The western blot in Figure 6B depicting the expression of the SRSF6 variants suggests that the phospho mimetic SRSF6 S303D is not expressed yet they claim this phosphorylation event determines the ratios of BAX and BAX-κ to fine tune the sensitivity to apoptosis. This western blot is not convincing. It might also be worthwhile to include an inducible protein control such as GFP (Figure 6D) which would not be expected to alter the ratio of Bax-kto Bax.

That original western blot was not the best. We have included another repeat of the western to better illustrate that SRSF6 S303D mutant is indeed expressed upon induction.